# COORDINATE IN AND VALUE OUT: TRAINING FLOW TRANSFORMERS IN AMBIENT SPACE

## ABSTRACT

Flow matching models have emerged as a powerful method for generative modeling on domains like images or videos, and even on unstructured data like 3D point clouds. These models are commonly trained in two stages: first, a data compressor (*i.e.* a variational auto-encoder) is trained, and in a subsequent training stage a flow matching generative model is trained in the low-dimensional latent space of the data compressor. This two stage paradigm adds complexity to the overall training recipe and sets obstacles for unifying models across data domains, as specific data compressors are used for different data modalities. To this end, we introduce **Ambient Space Flow Transformers** (ASFT), a domain-agnostic approach to learn flow matching transformers in ambient space, sidestepping the requirement of training compressors and simplifying the training process. We introduce a conditionally independent point-wise training objective that enables ASFT to make predictions continuously in coordinate space. Our empirical results demonstrate that using general purpose transformer blocks, ASFT effectively handles different data modalities such as images and 3D point clouds, achieving strong performance in both domains and outperforming comparable approaches. ASFT is a promising step towards domain-agnostic flow matching generative models that can be trivially adopted in different data domains.

## 1 INTRODUCTION

Recent advances in generative modeling have enabled learning complex data distributions by combining both powerful architectures and training objectives. In particular, state-of-the-art approaches for image (Esser et al., 2024), video (Dai et al., 2023) or 3D point cloud (Vahdat et al., 2022) generation are based on the concept of iteratively transforming data into Gaussian noise. Diffusion models were originally proposed following this idea and pushing the quality of generated samples in many different domains, including images (Dai et al., 2023; Rombach et al., 2022), 3D point clouds (Luo & Hu, 2021), graphs (Hoogeboom et al., 2022) and video (Ho et al., 2022a). More recently, flow matching (Lipman et al., 2023) and stochastic interpolants (Ma et al., 2024) have been proposed as generalized formulations of the noising process, moving from stochastic gaussian diffusion processes to general paths connecting a base (*e.g.* Gaussian) and a target (*e.g.* data) distribution.

In practice, these iterative refinement approaches are commonly applied in a low-dimensional *latent space* obtained from a pre-trained compressor model. Therefore, the training process for these approaches is composed of two independent training stages: in the first stage, a compressor (*e.g.* VAE (Vahdat et al., 2022), VQVAE (Ramesh et al., 2022), VQGAN (Rombach et al., 2022)) model is trained, using architectures that are specific to the data domain (*i.e.* ConvNets for image data (Rombach et al., 2022), PointNet for point clouds (Vahdat et al., 2022), etc. ) enforcing a bottleneck on the data dimensionality, with the goal of reducing compute cost of training the subsequent stage. In the second stage, general purpose transformer architectures are used for the generative modeling step (Peebles & Xie, 2023; Ma et al., 2024; Esser et al., 2024), where the distribution of latents is learnt. This type of generative modeling in latent space has become popular in the community due to its computational efficiency benefits obtained from compressed data dimensionality.

However, latent space generative modeling is not without drawbacks. An obvious shortcoming is that latent space generative models cannot benefit from end-to-end optimization, as data compressors and the downstream generative models are trained separately. In particular, two stage approaches are more

complex to implement than single stage models and involve tuning several hyper-parameters that can have a big impact on final performance (Rombach et al., 2022): spatial reduction ratio, adversarial loss weights or KL terms in VAEs (Rombach et al., 2022) . As an illustrative example, setting a high KL weight makes the problem of learning a distribution of latents trivial, yet results in very poor generation results. A very small KL weight on the other hand allows for great reconstruction performance for the first stage but fails to induce a suitable latent space for generative modeling (*e.g.* a dirac delta for each training sample in latent space). Our goal in this paper is to provide a single training stage approach that is domain-agnostic and simple to implement in practice, thus dispensing with the complexities of two stage training recipes and enabling modeling of different data modalities in ambient (*i.e.* data) space.

It is worth noting that training diffusion or flow matching models in ambient space is indeed possible when using domain specific architecture designs and training recipes. In the image domain, approaches have exploited its dense nature and applied cascaded U-Nets Ho et al. (2021; 2022b), joint training of U-Nets at multiple resolutions Gu et al. (2023), multi-scale losses (Hoogeboom et al., 2023) or U-Net transformer hybrids architectures (Crowson et al., 2024), obtaining strong results. However, developing strong domain-agnostic models, using general purposes architectures that can be applied across different data domains remains an important open problem.

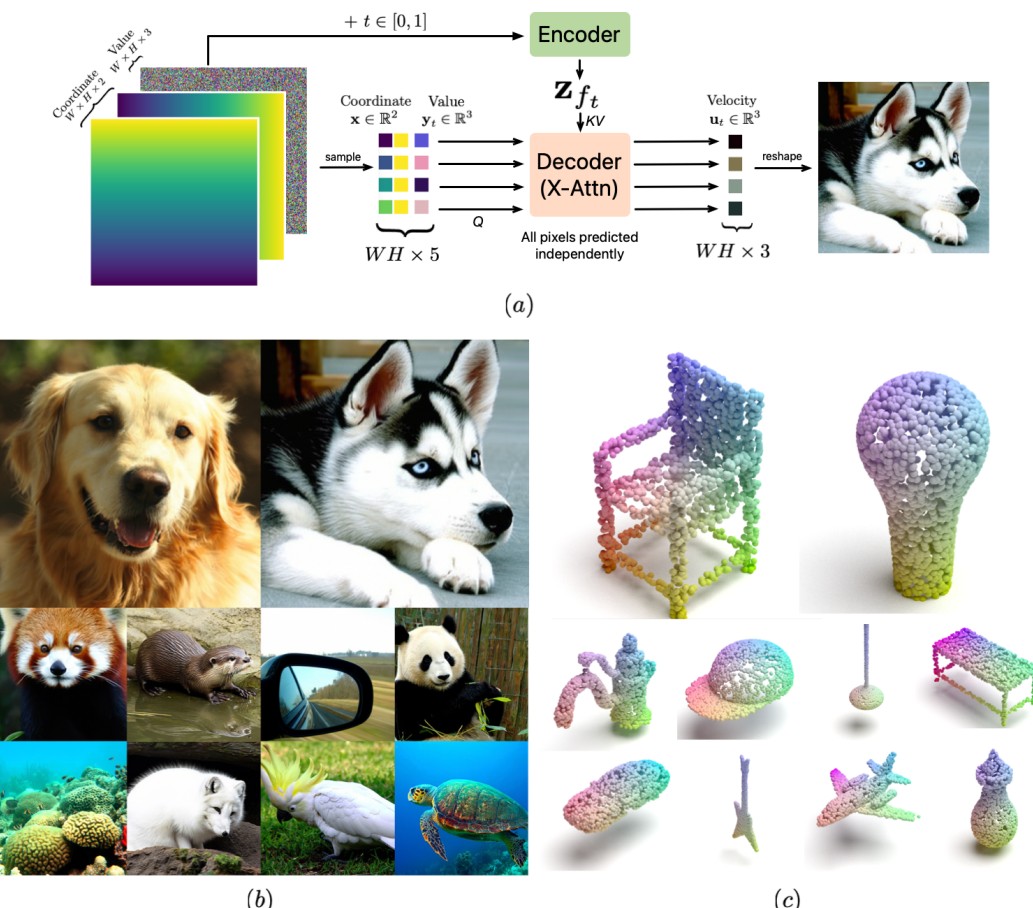

Figure 1: (a) High level overview of ASFT using the image domain as an example. Our model can be interpreted as an encoder-decoder model where the decoder makes predictions independently for each coordinate-value pair given $z_{f_t}$. For 3D point clouds, the coordinate and value are equivalent and their dimensions change, but the model is the same. (b) Samples generated by ASFT trained on ImageNet 256×256 and (c) 3D point clouds (2048 points) generated by training ASFT on ShapeNet.

In this paper, we answer a three part question: *Can we learn flow matching models in **ambient** space, in a **single training stage** and using a **domain agnostic** architecture?* Our goal is to unify different data domains under the same training recipe. To achieve this, we introduce Ambient Space

Flow Transformers (ASFT), see Fig. 1(a). ASFT makes progress towards the goal of unifying flow matching generative modeling across data domains. The key component of our approach is a conditionally independent point-wise training objective that enables training in ambient space and can be densely (*e.g.* continuously) evaluated during inference. In the image domain, this means that we model the probability of a pixel value given its coordinate, which provides granular and precise control over image synthesis, allowing to generate images at different resolution than the one used during training (see Fig. 4(a)). We show ImageNet-256 generated samples from ASFT in Fig. 1(b) and 3D point clouds from ShapeNet in Fig. 1(c) (see additional samples in Fig. 6, 7, 8, 9). Our contributions are summarized as follows:

- We propose ASFT, a flow matching generative transformer that works on ambient space to enable single stage generative modeling on different data domains.

- Our results show that ASFT, though domain-agnostic, achieves competitive performance on image and 3D point cloud generation compared with strong domain-specific baselines.

- Our point-wise training objective allows for efficient training via sub-sampling dense domains like images while also enabling resolution changes at inference time.

## 2 RELATED WORK

Diffusion models have been the major catalyzer of progress in generative modeling, these approaches learn to reverse a forward process that gradually adds Gaussian noise to corrupt data samples (Ho et al., 2020). Diffusion models are notable for their simple and robust training objective. Extensive research has explored various formulations of the forward and backward processes (Song et al., 2021a; Rissanen et al., 2022; Bansal et al., 2022), particularly in the image domain. In addition, different denoising networks have been proposed for different data domains like images (Nichol & Dhariwal, 2021), videos (Ho et al., 2022a), and geometric data (Luo & Hu, 2021). More recently, flow matching (Liu et al., 2023; Lipman et al., 2023) and stochastic interpolants (Ma et al., 2024) have emerged as flexible formulations that generalized Gaussian diffusion paths, allowing to define different paths to connect a base and a target distribution. These types of models have shown incredible results in the image domain (Ma et al., 2024; Esser et al., 2024) when coupled with transformer architectures (Vaswani et al., 2017) to model distributions in latent space learnt by data compressors (Peebles & Xie, 2023; Ma et al., 2024; Rombach et al., 2022; Vahdat et al., 2022; Zheng et al., 2023; Gao et al., 2023). Note that these approaches train two separate stages/models: first training the data compressor (*e.g.* VAE (Vahdat et al., 2022), VQVAE (Ramesh et al., 2022), VQGAN (Rombach et al., 2022)) and then the generative model, requiring careful hyper-parameter tuning.

In an attempt to unify generative modeling across various data domains, continuous data representations[1] have shown potential in different approaches: From Data to Functa (Functa) (Dupont et al., 2022a), Generative Manifold Learning (GEM) (Du et al., 2021a), and Generative Adversarial Stochastic Process (GASP) (Dupont et al., 2022b) have studied the problem of generating continuous representations of data. More recently Infinite Diffusion (Bond-Taylor & Willcocks, 2023) and PolyINR (Singh et al., 2023) have shown great results in the image domain by modeling images as continuous functions. However, both of these approaches make strong assumptions about image data. In particular, (Bond-Taylor & Willcocks, 2023) interpolates sparse pixels to an euclidean grid to then process it with a U-Net. On the other hand, (Singh et al., 2023) uses a patching and 2D convolution in the discriminator. Our approach also relates to DPF Zhuang et al. (2023), a diffusion model that acts on function coordinates and can be applied in different data domains on a grid at low resolutions (*i.e.* 64×64). Our approach is able to deal with higher resolution functions (*e.g.* 256x256 vs. 64x64 resolution images) on large scale datasets like ImageNet, while also tackling unstructured data domains that do not live on an Euclidean grid (*e.g.* like 3D point clouds).

---

[1]These are also often referred to as *implicit neural representation*, *neural maps* or *neural operators*

## 3 METHOD

### 3.1 DATA AS COORDINATE → VALUE MAPS

We interpret our empirical data distribution $q$ to be composed of maps $f \sim q(f)$. These maps take *coordinates $x$* as input to *values $y$* as output. For images, maps are defined from 2D pixel coordinates $x \in \mathbb{R}^2$ to corresponding RGB values $y \in \mathbb{R}^3$, thus $f : \mathbb{R}^2 \to \mathbb{R}^3$, where each image is a different map. For 3D point clouds, $f$ can be interpreted as a deformation that maps coordinates from a fixed base configuration in 3D space to a deformation value also in 3D space, $f : \mathbb{R}^3 \to \mathbb{R}^3$, as in the image case, each 3D point cloud corresponds to a different deformation map $f$. For ease of notation, we define coordinates $x$ and values $y$ of any given map $f$ as $x_f$ and $y_f$, respectively. Fig. 1(a) shows an example of such maps in the image domain.

In practice, analytical forms for these maps $f$ are unknown. In addition, different from previous approaches (Dupont et al., 2022a; Du et al., 2021a), we do not assume that parametric forms of these maps can be obtained, since that would involve a separate training stage fitting an MLP to each map (Dupont et al., 2022a; Bauer et al., 2023; Du et al., 2021a). As a result, we assume we are only given sets of corresponding *coordinate* and *value* pairs resulting from observing these maps at a particular sampling rate (*e.g.* at a particular resolution in the image case). Therefore, we need a to develop an end-to-end approach that can take these collections of coordinate-value sets as training data.

### 3.2 FLOW MATCHING AND STOCHASTIC INTERPOLANTS

We consider generative models that learn to reverse a time-dependent forward process that turns data samples (*i.e.* maps $f$ in our case) $f \sim q(f)$ into noise $\epsilon \sim \mathcal{N}(0, \mathbf{I})$.

$$f_t = \alpha_t f + \sigma_t \epsilon \tag{1}$$

Both flow matching (Lipman et al., 2023) and stochastic interpolant (Ma et al., 2024) formulations build this forward process in Eq. 1 so that it interpolates exactly between data samples $f$ at time $t = 0$ and $\epsilon$ at time $t = 1$, with $t \in [0, 1]$. In particular, $p_1(f) \sim \mathcal{N}(0, \mathbf{I})$ and $p_0(f) \approx q(f)$. In this case, the marginal probability distribution $p_t(f)$ of $f$ is equivalent to the distribution of the probability flow ODE with the following velocity field (Ma et al., 2024):

$$d_t f_t = \boldsymbol{u}_t(f_t) d_t \tag{2}$$

where the velocity field is given by the following conditional expectation,

$$\boldsymbol{u}_t(f) = \mathbb{E}[d_t f_t | f_t = f] = d_t \alpha_t \mathbb{E}[f_0 | f_t = f] + d_t \sigma_t \mathbb{E}[\epsilon | f_t = f]. \tag{3}$$

Under this formulation, samples $f_0 \sim p_0(f)$ are generated by solving the probability flow ODE in Eq. 2 backwards in time (*e.g.* . flowing from $t = 1$ to $t = 0$), where $p_0(f) \approx q(f)$. Note that both the flow matching (Lipman et al., 2023) and stochastic interpolant (Ma et al., 2024) formulations decouple the time-dependent process formulation from the specific choice of parameters $\alpha_t$ and $\sigma_t$, allowing for more flexibility. Throughout the presentation of our method we will assume a rectified flow (Liu et al., 2023; Lipman et al., 2023) or linear interpolant path (Ma et al., 2024) between noise and data, which define a straight path to connect data and noise: $f_t = (1 - t) f_0 + t\epsilon$. Note that our framework for learning flow matching models for coordinate-value sets can be used with any path definition. Compared with diffusion models (Ho et al., 2020), linear flow matching objectives result in better training stability and more modeling flexibility (Ma et al., 2024; Esser et al., 2024) which we observed in our early experiments.

### 3.3 FLOW MATCHING FOR COORDINATE-VALUE SETS

We now turn to the task of formulating a flow matching training objective for data distributions of maps $f$. We recall that in practice we do not have access to an analytical or parametric form for these maps $f$, and we are only given sets of corresponding *coordinate $x_f$* and *value $y_f$* pairs resulting from

observing the mapping at a particular rate. As a result, we need to formulate a training objective that can take these sets of coordinate-value as training data.

In order to achieve this, we first observe that the target velocity field $\boldsymbol{u}_t(f_t)d_t$ can be decomposed across both the domain and co-domain of $f_t$, resulting in a *point-wise velocity field* $\boldsymbol{u}_t(\boldsymbol{x}_{f_t}, \boldsymbol{y}_{f_t})d_t$, defined for corresponding coordinate and value pairs of $f_t$. As an illustrative example in the image domain, this means that the *target velocity field can be independently evaluated* for any pixel coordinate $\boldsymbol{x}_{f_t}$ with corresponding value $\boldsymbol{y}_{f_t}$, so that $\boldsymbol{u}_t(\boldsymbol{x}_{f_t}, \boldsymbol{y}_{f_t}) \in \mathbb{R}^3$. Note that one can always decompose target velocity fields in this way since the time-dependent forward process in Eq. 1 aggregates data and noise *independently* (*e.g.* point-wise) across the domain of $f$. Again, using the image domain as an example, the time-dependent forward process of a pixel at coordinate $\boldsymbol{x}_f$ is not dependent on other pixel positions or values.

Our goal now is to formulate a training objective to match this point-wise independent velocity field. We want our neural network $\boldsymbol{v}_\theta$ parametrizing the velocity field to be able to independently predict a velocity for any given coordinate and value pair $\boldsymbol{x}_{f_t}$ and $\boldsymbol{y}_{f_t}$. However, this point-wise independent prediction is futile without access to additional contextual conditioning information about the underlying function $f_t$ at time $t$. This is because even if the forward process is point-wise independent, real data exhibits strong dependencies across the domain $f$ that need to be captured by the model. For example, in the image domain, pixels are not independent from each other and natural images show strong both short and long spatial dependencies across pixels. In order to solve this, we introduce a latent variable $\boldsymbol{z}_{f_t}$ that encodes contextual information from a set of given coordinate and value pairs of $f_t$. This contextual latent variable allows us to formulate the learnt velocity field to be *conditionally independent* for coordinate-value pairs given $\boldsymbol{z}_{f_t}$. The final point-wise conditionally independent CFM loss, which we denote as CICFM loss is defined as:

$$L_{\text{CICFM}} = \mathbb{E}_{t \sim \mathcal{U}[0,1], f \sim q(f), \epsilon \sim \mathcal{N}(0,\mathbf{I})} ||\boldsymbol{v}_\theta(\boldsymbol{x}_{f_t}, \boldsymbol{y}_{f_t}, t|\boldsymbol{z}_{f_t}) - \boldsymbol{u}_t(\boldsymbol{x}_f, \boldsymbol{y}_f|\epsilon)||_2^2, \qquad (4)$$

where the target velocity field $\boldsymbol{u}_t(\boldsymbol{x}, \boldsymbol{y}|\epsilon)$ is defined as a rectified flow (Liu et al., 2023; Lipman et al., 2023) or linear interpolant path (Ma et al., 2024):

$$\boldsymbol{u}_t(\boldsymbol{x}_f, \boldsymbol{y}_f|\epsilon) = (1-t)\epsilon + t\boldsymbol{y}_f. \qquad (5)$$

One of the core challenges of learning this type of generative models is obtaining a latent variable $\boldsymbol{z}_{f_t}$ that effectively captures intricate dependencies across the domain of the function, specially for high resolution stimuli like images. In particular, the architectural design decisions are extremely important to ensure that $\boldsymbol{z}_{f_t}$ does not become a bottleneck during training. In the following we review our proposed architecture.

## 3.4 NETWORK ARCHITECTURE

We base our model on the general PerceiverIO design (Jaegle et al., 2022) which provides a flexible architecture to handle coordinate-value sets of large cardinality (*i.e.* large number of pixels in an image). Fig. 2 illustrates the architectural pipeline of ASFT. At a high level, our encoder network takes a set of coordinate-value pairs and encodes them to learnable latents through cross-attention. These latents are then updated through several self-attention blocks to provide the final latents $\boldsymbol{z}_{f_t} \in \mathbb{R}^{L \times D}$. To decode the velocity field for a given coordinate-value pair we perform cross attention to $\boldsymbol{z}_{f_t}$, generating the final point-wise prediction for the velocity field $\boldsymbol{v}_\theta(\boldsymbol{x}_{f_t}, \boldsymbol{y}_{f_t}, t|\boldsymbol{z}_{f_t})$.

The encoder of a vanilla PerceiverIO relies solely on cross-attention to the latents $z_{f_t} \in \mathbb{R}^{L \times D}$ to learn spatial connectivity patterns between input and output elements, which we found to introduce a strong bottleneck during training. To ameliorate this, we make two key modifications to boost the performance. Firstly, our encoder utilizes spatial aware latents where each latent is assigned a "pseudo" coordinate. Coordinate-value pairs are assigned to different latents based on their distances on coordinate space. During encoding, coordinate-value pairs interact with their assigned latents through cross-attention. In particular, the learnable latent $z_{f_t}$ cross-attends to input coordinate-value pairs of noisy data at a given timestep $t$. Latent vectors are spatial-aware, this means that each of the $L$ latents only attends to a set of neighboring coordinate-value pairs. Latent vectors are then updated using several self-attention blocks. These changes in the encoder allow the model to effectively utilize

Figure 2: Architecture of our proposed ASFT for different data domains including images and 3D point clouds. Note that models are trained for each data domain separately. Each spatial aware latent takes in a subset of neighboring context coordinate-value sets in coordinate space. The latents are then updated through self-attention. Decoded coordinate-value pairs cross attend to the updated latents $z_{f_t}$ to decode the corresponding velocity.

spatial information while also saving compute when encoding large coordinate-value sets on ambient space. In the decoder, a given coordinate-value pair cross attends to $z_{f_t}$ as in the original PerceiverIO. However, we found that a multi-level decoding strategy, which not only cross attends to the latents in the final layer but also the latent from the intermediate self-attentions layer is helpful. In particular, a given coordinate-value pair cross attends to latents from subsequent encoder layers sequentially to progressively refine the prediction. Finally, following previous work (Peebles & Xie, 2023; Ma et al., 2024), we apply AdaLN-Zero blocks for conditioning both on timestep $t$ and class labels whenever needed (*e.g.* for ImageNet experiments). More architectural details can be found in App. A.

## 4 EXPERIMENTS

We evaluate ASFT on two challenging problems: image generation (FFHQ-256 (Karras et al., 2019), LSUN-Church-256 (Yu et al., 2015), ImageNet-128/256 (Russakovsky et al., 2015)) and 3D point cloud generation (ShapeNet (Chang et al., 2015)). Note that we use the same training recipe both tasks, adapted for changes in coordinate-value pair dimensions in different domains. See App. A for more implementation details and training settings.

ASFT enables practitioners to define the number of coordinate-value pairs to be decoded during training. In our experiments, we set the number of decoded coordinate-value pairs to 4096 for images with resolution $128 \times 128$, 8192 for images with resolution $256 \times 256$, and 2048 for point clouds unless mentioned otherwise. On image generation, we train models with small (S), base (B), large (L), and extra large (XL) sizes. For 3D point cloud generation we set the parameter count to match the model size in previous state-of-the-art approaches (*i.e.* LION (Vahdat et al., 2022)). Detailed configuration for all models can be found in Appendix A. During inference, we adopt black-box numerical ODE solver with maximal NFE as 100 for image generation (Song et al., 2021b) and an SDE sampler with 1000 steps for point cloud generation to match the settings in (Vahdat et al., 2022).

### 4.1 IMAGE GENERATION IN FUNCTION SPACE

Given that ASFT is a generative model for maps we compare it with other generative models of the same type, namely approaches that operate in function spaces. Tab. 1 shows a comparison of different image domain specific as well as function space models (*e.g.* approaches that model infinite-dimensional signals). ASFT surpasses other generative models in function space on both FFHQ (Karras et al., 2019) and LSUN-Church (Yu et al., 2015) at resolution $256 \times 256$. Compared with generative models designed specifically for images, ASFT also achieves comparable or better performance. When scaling up the model size, ASFT-L demonstrates better performance than all the

| Model | FFHQ-256 | Church-256 |
|---|---|---|
| *Domain specific models* | | |
| CIPS (Anokhin et al., 2021) | 5.29 | 10.80 |
| StyleSwin (Zhang et al., 2022) | 3.25 | 8.28 |
| UT (Bond-Taylor et al., 2022) | 3.05 | 5.52 |
| StyleGAN2 (Karras et al., 2020) | 2.35 | 6.21 |
| *Function space models* | | |
| GEM (Du et al., 2021b) | 35.62 | 87.57 |
| GASP (Dupont et al., 2022c) | 24.37 | 37.46 |
| ∞-Diff (Bond-Taylor & Willcocks, 2023) | 3.87 | 10.36 |
| **ASFT**-B (ours) | 2.46 | 7.11 |
| **ASFT**-L (ours) | **2.18** | **5.51** |

Table 1: $\text{FID}_{\text{CLIP}}$ (Kynkäänniemi et al., 2023) results for state-of-the-art function space approaches.

baselines on FFHQ-256 and Church-256, indicating that ASFT can benefit from increasing model sizes.

## 4.2 IMAGENET

We also evaluate the performance of ASFT on large scale and challenging settings, we train ASFT on ImageNet at both 128×128 and 256×256 resolutions. On ImageNet-128, shown in Tab. 2, ASFT achieves an FID of 2.73, which is a a competitive performance in comparison to diffusion or flow-based generative baselines including ADM (Dhariwal & Nichol, 2021), CDM (Ho et al., 2021), and RIN (Jabri et al., 2023) which use domain-specific architectures for image generation. Besides, comparing to PolyINR (Singh et al., 2023) which also operates on function space, ASFT achieves competitive FID, while obtaining better IS, precision and recall. The experimental results demonstrate the capabilities of ASFT in generating realistic samples on large scale datasets.

We report results of ASFT for ImageNet-256 on Tab. 3. Note that ASFT is slightly outperformed by latent space models like DiT (Peebles & Xie, 2023) and SiT (Ma et al., 2024). We highlight that these baselines rely on a pre-trained VAE compressor that was trained on datasets that are much larger than ImageNet, while ASFT was trained only with ImageNet data. In addition, ASFT achieves better performance than many of the baselines trained only with ImageNet data including ADM (Dhariwal & Nichol, 2021), CDM (Ho et al., 2021) and Simple Diffusion (U-Net) (Hoogeboom et al., 2023) which all use CNN-based architectures specific for image generation. Note that this is consistent with the results show in Tab. 1, where ASFT outperforms all function space approaches. When comparing with approaches using transformers architectures we find that ASFT obtains performance comparable to RIN (Jabri et al., 2022) and HDiT (Crowson et al., 2024), with slightly worse FID and slightly better IS. However, ASFT is a domain-agnostic architecture that can be trivially applied to different data domains like 3D point clouds (see Sect. 4.3). For completeness, we also include a comparison with very large U-Net transformer hybrid models, Simple Diffusion (U-ViT 2B) and VDM++ (U-ViT 2B) which both use approx. ×2.72 more parameters than ASFT-XL, unsurprisingly, these much bigger capacity models outperform ASFT (see App. A for a more detailed comparison including training settings). We highlight that the simplicity of implementing and training ASFT models in practice, and the trivial extension to different data domains (as shown in Sect. 4.3) are strong arguments favouring our model. Finally, comparing with *e.g.* PolyINR (Singh et al., 2023) which is also a function space generative model we also find comparable performance, with slight worse FID but better Precision and Recall. It is worth noting that (Singh et al., 2023) applies a pre-trained DeiT model as the discriminator (Singh et al., 2023). Whereas our ASFT makes no such assumption about the function or pre-trained models, enabling to trivially apply ASFT to other domains like 3D point clouds (see Sect. 4.3).

To demonstrate the scalability of ASFT we train models of different sizes including small (S), base (B), large (L), and extra-large (XL) on ImageNet-256. We show the performance of different model sizes using FID-50K in Fig. 3(a). We observe a clear improving trend when increasing the number of parameters as well as increasing training steps. This demonstrates that scaling the total training Gflops is important to improved generative results as in other ViT-based generative models (Peebles & Xie, 2023; Ma et al., 2024). Due to the flexibility of cross-attention decoder in ASFT, one can easily conduct random sub-sampling to reduce the number of decoded coordinate-value pairs during training which significantly saves computation. Fig. 3(b) shows how number of decoded coordinate-

| Class-Conditional ImageNet 128x128 | | | | |
|---|---|---|---|---|
| Model | FID↓ | IS↑ | Precision↑ | Recall↑ |
| *Adversarial models* | | | | |
| BigGAN-deep (Brock et al., 2019) | 6.02 | 145.8 | **0.86** | 0.35 |
| PolyINR (Singh et al., 2023) | **2.08** | 179.0 | 0.70 | 0.45 |
| *Diffusion models* | | | | |
| CDM (w/ cfg) (Ho et al., 2021) | 3.52 | 128.0 | - | - |
| ADM (w/ cfg) (Dhariwal & Nichol, 2021) | 2.97 | 141.3 | 0.78 | **0.59** |
| RIN (Jabri et al., 2023) | 2.75 | 144.0 | - | - |
| **ASFT**-XL (ours) (cfg=1.5) | 2.73 | **187.6** | 0.80 | 0.58 |

Table 2: Benchmarking class-conditional image generation on ImageNet 128x128.

| Class-Conditional ImageNet 256x256 | | | | | | | |
|---|---|---|---|---|---|---|---|
| Model | Agnostic | #Samples | #Params | FID↓ | IS↑ | Precision↑ | Recall↑ |
| *Adversarial models* | | | | | | | |
| BigGAN-deep (Brock et al., 2019) | ✗ | 1.28M | - | 6.95 | 171.4 | **0.87** | 0.28 |
| PolyINR (Singh et al., 2023) | ✗ | 1.28M | - | 2.86 | 241.4 | 0.71 | 0.39 |
| *Latent space with pretrained VAE* | | | | | | | |
| DiT-XL (cfg=1.5) (Peebles & Xie, 2023) | ✗ | 9.23M | 675M | 2.27 | **278.2** | 0.83 | 0.57 |
| SiT-XL (cfg=1.5, SDE) (Ma et al., 2024) | ✗ | 9.23M | 675M | **2.06** | 270.2 | 0.82 | 0.59 |
| *Ambient space* | | | | | | | |
| ADM (Dhariwal & Nichol, 2021) | ✗ | 1.28M | 554M | 10.94 | 100.9 | 0.69 | **0.63** |
| CDM (Ho et al., 2021) | ✗ | 1.28M | - | 4.88 | 158.7 | - | - |
| Simple Diff. (U-Net) (Hoogeboom et al., 2023) | ✗ | 1.28M | - | 3.76 | 171.6 | - | - |
| RIN (Jabri et al., 2023) | ✗ | 1.28M | 410M | 3.42 | 182.0 | - | - |
| HDiT (cfg=1.3) (Crowson et al., 2024) | ✗ | 1.28M | 557M | 3.21 | 220.6 | - | - |
| Simple Diff. (U-ViT) Hoogeboom et al. (2023) | ✗ | 1.28M | 2B | 2.77 | 211.8 | - | - |
| VDM++ (U-ViT) (Kingma & Gao, 2023) | ✗ | 1.28M | 2B | 2.12 | 267.7 | - | - |
| **ASFT**-XL (ours) (cfg=1.5) | ✓ | 1.28M | 733M | 3.74 | 228.8 | 0.82 | 0.52 |

Table 3: Top performing models for class-conditional image generation on ImageNet 256x256.

value pairs affects the model performance as well as Gflops in training. An image of resolution $256\times256$ contains 65536 pixels in total which is the maximal number of coordinate-value pairs during training. As see in Fig. 3(b), a model decoding 4096 coordinate-value pairs saves more than 20% Gflops over one decoding 16384. This provides us with an effective training recipe, which saves computation by only decoding a subset of 12% of the image pixels during training. Interestingly, we see a performance drop when densely decoding 16384 coordinate value pairs. We hypothesize this could be due to optimization challenges of decoding large numbers of pairs and leave further analysis for future work.

## 4.3 SHAPENET

To show the domain-agnostic prowess of ASFT we also tackle 3D point cloud generation on ShapeNet (Chang et al., 2015). Note that our model does not require training separate VAEs for point clouds, tuning their corresponding hyper-parameters or designing domain specific networks. We simply adapt our architecture for the change in dimensionality of coordinate-value pairs (*e.g.* $f : \mathbb{R}^2 \to \mathbb{R}^3$ for images to $f : \mathbb{R}^3 \to \mathbb{R}^3$ for 3D point clouds.). Note that for 3D point clouds, the coordinates and values are equivalent. In this setting, we compare baselines including LION (Vahdat et al., 2022) which is a recent state-of-the-art approach that models 3D point clouds using a latent diffusion type of approach. Following Vahdat et al. (2022) we report MMD, COV and 1-NNA as metrics. To have a straightforward comparison with baselines, we train ASFT-B with to approximately match the number of parameters as LION (Vahdat et al., 2022) (110M for LION vs 108M for ASFT) on the same datasets (using per sample normalization as in Tab. 17 in Vahdat et al. (2022)). We show results for category specific models and for an unconditional model jointly trained on 55 ShapeNet categories in Tab. 4. ASFT-B obtains strong generation results on ShapeNet despite being a domain agnostic approach and outperforms LION in most datasets and metrics. Note that ASFT-B has comparable number of parameters and the same inference settings than LION so this is fair

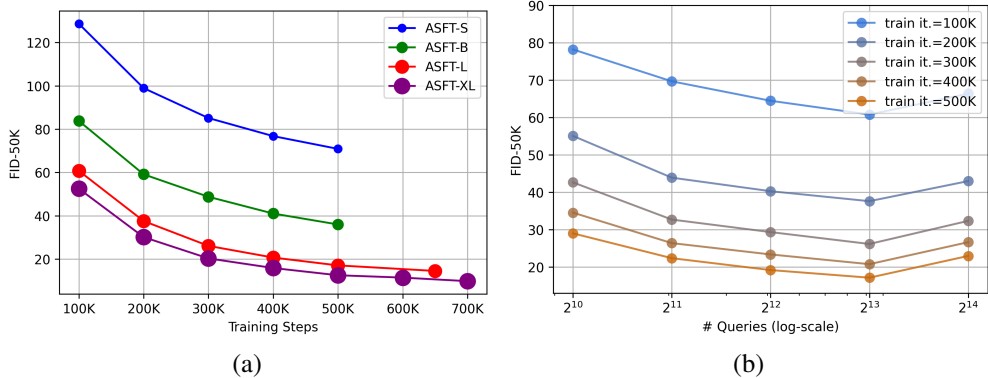

(a)  (b)

Figure 3: (a) FID-50K over training iterations with different model sizes, where we see clear benefits of scaling up model sizes. (b) FID-50K over training iterations with different number of decoded coordinate-value pairs during training and the corresponding compute cost for a single forward pass.

comparison. Finally, we also report results for a larger model ASFT-L (with ×2 the parameter count as LION) to investigate how ASFT improves as with increasing model size. We observe that with increasing model size, ASFT typically achieves better performance than the base version. This further demonstrates scalability of our model on ambient space of different data domains.

| Category | Model | MMD↓ | | COV↑ (%) | | 1-NNA↓ (%) | |
|---|---|---|---|---|---|---|---|
| | | CD | EMD | CD | EMD | CD | EMD |
| Airplane | ShapeGF (Cai et al., 2020) | 0.3130 | 0.6365 | **45.19** | 40.25 | 81.23 | 80.86 |
| | SP-GAN (Li et al., 2021) | 0.4035 | 0.7658 | 26.42 | 24.44 | 94.69 | 93.95 |
| | GCA (Zhang et al., 2021) | 0.3586 | 0.7651 | 38.02 | 36.30 | 88.15 | 85.93 |
| | LION (Vahdat et al., 2022) (110M) | 0.3564 | 0.5935 | 42.96 | **47.90** | 76.30 | 67.04 |
| | **ASFT**-B (ours) (108M) | **0.2861** | 0.5156 | 43.38 | 47.54 | 75.55 | 64.95 |
| | **ASFT**-L (ours) | 0.2880 | **0.5052** | 44.44 | 47.16 | **62.20** | 62.96 |
| Chair | ShapeGF (Cai et al., 2020) | 3.7243 | 2.3944 | 48.34 | 44.26 | 58.01 | 61.25 |
| | SP-GAN (Li et al., 2021) | 4.2084 | 2.6202 | 40.03 | 32.93 | 72.58 | 83.69 |
| | GCA (Zhang et al., 2021) | 4.4035 | 2.5820 | 45.92 | 47.89 | 64.27 | 64.50 |
| | LION (Vahdat et al., 2022) (110M) | 3.8458 | 2.3086 | 46.37 | 50.15 | 56.50 | 53.85 |
| | **ASFT**-B (ours) (108M) | 3.6310 | **2.1725** | 46.67 | **53.31** | 55.43 | **51.13** |
| | **ASFT**-L (ours) | **3.5145** | 2.1860 | **49.39** | 49.84 | **50.52** | 51.66 |
| Car | ShapeGF (Cai et al., 2020) | 1.0200 | 0.8239 | **44.03** | 47.16 | 61.79 | 57.24 |
| | SP-GAN (Li et al., 2021) | 1.1676 | 1.0211 | 34.94 | 31.82 | 87.36 | 85.94 |
| | GCA (Zhang et al., 2021) | 1.0744 | 0.8666 | 42.05 | 48.58 | 70.45 | 64.20 |
| | LION (Vahdat et al., 2022) (110M) | 1.0635 | 0.8075 | 42.90 | **50.85** | 59.52 | **49.29** |
| | **ASFT**-B (ours) (108M) | 0.9923 | **0.7692** | 43.46 | 47.44 | 60.36 | 53.27 |
| | **ASFT**-L (ours) | **0.9660** | 0.7846 | **44.03** | 48.86 | 53.83 | 54.55 |
| All (55 cat) | LION (Vahdat et al., 2022) (110M) | 3.4336 | 2.0953 | 48.00 | 52.20 | 58.25 | 57.75 |
| | **ASFT**-B (ours) (108M) | 3.2586 | 2.1328 | 49.00 | 50.40 | 54.65 | 55.70 |
| | **ASFT**-L (ours) | **3.1775** | **1.9794** | **49.80** | **52.39** | **51.80** | **53.90** |

Table 4: Generation performance metrics on Airplane, Chair, Car and all 55 categories jointly. All models were trained on the ShapeNet dataset from PointFlow (Yang et al., 2019). Both the training and testing data are normalized individually into range [-1, 1].

## 4.4 RESOLUTION AGNOSTIC GENERATION

An interesting property of ASFT is that it decodes each coordinate-value pair independently, allowing resolution to change during inference. At inference the user can define as many coordinate-value pairs as desired where the initial value of each pair at $t = 1$ is drawn from a Gaussian distribution. We show qualitative results of resolution agnostic generation for both images and point clouds. Fig. 4(a) show images sampled at resolution $512 \times 512$ (together with their 256 resolution counterparts generated from the same seed) from ASFT trained on ImageNet-256. Even though the model has not been trained with any samples at 512 resolution, it can still generate realistic images with high-frequency details. Fig. 4(b) shows point cloud with 100K points from ASFT trained on ShapeNet with only

2048 points points per sample (we visualize the generated 2048 point could generated from the same seed). Similarly, ASFT generates dense and realistic point cloud in 3D without actually being trained on such high density points. These results show that ASFT is not trivially overfitting to the training set of points but rather learning a continuous density field in 3D space from which an infinite number of points could be sampled. Generally speaking, this also provides the potential to efficiently train flow matching generative models without the need to use large amounts of expensive high resolution data, which can be hard to collect in data domains other than images.

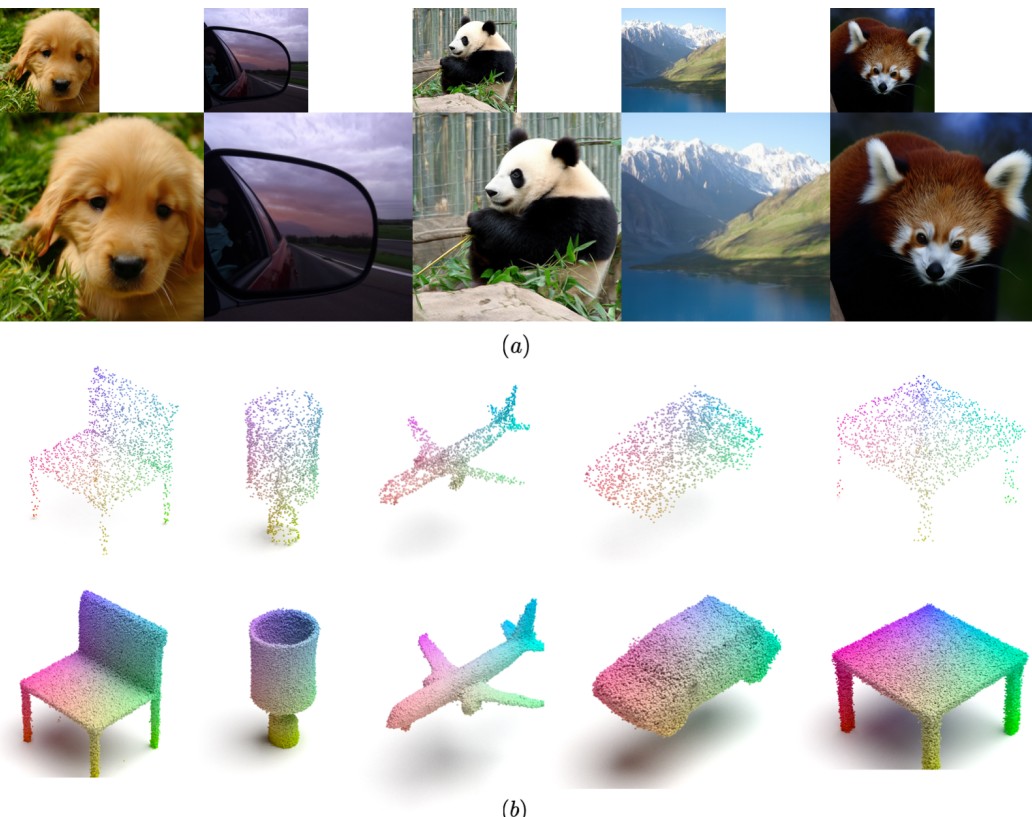

Figure 4: (a) **Top:** images generated at 256 resolution from an ASFT trained on ImageNet-256. **Bottom:** Samples generated by ASFT from the same seed at 512 resolution . (b) **Top:** Point clouds generated by ASFT containing 2048 points each. **Bottom:** Samples generated by ASFT from the same seed containing 100K points each, $50\times$ more points than seen during training.

## 5 CONCLUSION

We introduced Ambient Space Flow Transformers (ASFT), a flow matching generative model designed to operate directly in ambient space. Our approach dispenses with the practical complexities of training latent space generative models, such as the dependence on domain-specific compressors for different data domains or tuning of hyper-parameters of the data compressor (*i.e.* adversarial weight, KL term, etc.). We introduced a conditionally independent point-wise training objective that decomposes the target vector field and allows to continuously evaluate the generated samples, enabling resolution changes at inference time. This training objective also improves training efficiency since it allows us to sub-sample the target vector field during training. Our results on both image and 3D point cloud benchmarks show the strong performance of ASFT as well as its trivial adaption across modalities. In conclusion, ASFT represents a promising direction for flow matching generative models, offering a powerful and domain-agnostic framework. Future work could explore further improvements in training efficiency and investigate co-training of multiple data domains to enable multi-modality generation in an end-to-end learning paradigm.

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

# A  MODEL CONFIGURATION AND TRAINING SETTINGS

We provide detailed model configurations and training settings of ASFT for image (Tab. 5) and point cloud (Tab. 6) generation. For image generation, we develop model sizes small (S), base (B), large (L), and extra large (XL) to approximately match the number of parameters in previous works (Peebles & Xie, 2023). Similarly, for point cloud generation, we train a base sized model roughly matching the number of parameters in LION (Vahdat et al., 2022) (*i.e.* 110M parameters), and a ASFT-L which contains about twice the number of parameters as ASFT-B. For image experiments we implement the "psuedo" coordinate of latents as 2D grids and coordinate-value pairs are assigned to different latents based on their distances to the latent coordinates. Whereas in point cloud generation, since calculating the pair-wise distances in 3D space can be time consuming, we assign input elements to latents through a hash code, so that neighboring input elements are likely (but not certainly) to be assigned to the same latent token. We found that the improvements of spatial aware latents in 3D to not be as substantial as in the 2D image setting, so we report results with a vanilla PerceiverIO architecture for simplicity. To embed coordinates, we apply standard Fourier positional embedding (Vaswani et al., 2017) for ambient space coordinate input in both encoder and decoder. The Fourier positional embedding is also applied to the "psuedo" coordinate of latents. On image generation, we found that applying rotary positional embedding (RoPE) (Su et al., 2024) slightly improves the performance of ASFT. Therefore, RoPE is employed for largest ASFT-XL model. For all the models including image and 3D point cloud experiments, we share the following training parameters except the `training_steps` across different experiments. On image generation, all models are trained with batch size 256, except for ASFT-XL reported in Tab. 2 and Tab. 3, which are trained for 1.7M steps with batch size 512. On ShapeNet, ASFT models are trained for 800K iterations with a batch size of 16.

```
default training config:
    optimizer='AdamW'
    adam_beta1=0.9
    adam_beta2=0.999
    adam_eps=1e-8
    learning_rate=1e-4
    weight_decay=0.0
    gradient_clip_norm=2.0
    ema_decay=0.999
    mixed_precision_training=bf16
```

In Tab. 7, we also compare the size of models trained on ImageNet-256, training cost (*i.e.* product of batch size and training iterations), and inference cost (*i.e.* NFE, number of function evaluation). Note that for models that achieve better performance than ASFT, many of them are trained for more iterations. In addition, at inference time ASFT applies simple first order Euler sampler with 100 sampling steps, which uses less NFE than many other baselines.

| Model | Layers | Hidden size | #Latents | Heads | Decoder layers | #Params |
|---|---|---|---|---|---|---|
| ASFT-S | 12 | 384 | 1024 | 6 | 1 | 35M |
| ASFT-B | 12 | 768 | 1024 | 12 | 1 | 138M |
| ASFT-L | 24 | 1024 | 1024 | 16 | 1 | 458M |
| ASFT-XL | 28 | 1152 | 1024 | 16 | 2 | 733M |

Table 5: Detailed configurations of ASFT for image generation.

| Model | Layers | Hidden size | #Latents | Heads | Decoder layers | #Params |
|---|---|---|---|---|---|---|
| ASFT-B | 9 | 512 | 1024 | 4 | 1 | 108M |
| ASFT-L | 12 | 512 | 1024 | 4 | 1 | 204M |

Table 6: Detailed configurations of ASFT for point cloud generation.

| Model | # Train data | # params | bs.×it. | NFE | FID ↓ | IS ↑ |
|---|---|---|---|---|---|---|
| ADM (Dhariwal & Nichol, 2021) | 1.28M | 554M | 507M | 1000 | 10.94 | 100.9 |
| RIN (Jabri et al., 2023) | 1.28M | 410M | 614M | 1000 | 3.42 | 182.0 |
| HDiT (Crowson et al., 2024) | 1.28M | 557M | 742M | 100 | 3.21 | 220.6 |
| Simple Diff. (U-ViT 2B) (Hoogeboom et al., 2023) | 1.28M | 2B | 1B | - | 2.77 | 211.8 |
| DiT-XL (Peebles & Xie, 2023) | 9.23M | 675M | 1.8B | 250 | 2.27 | 278.2 |
| VDM++ (U-ViT 2B) (Kingma & Gao, 2023) | 1.28M | 2B | 1.4B | 512 | 2.12 | 267.7 |
| SiT-XL (Ma et al., 2024) | 9.23M | 675M | 1.8B | 500 | 2.06 | 270.2 |
| ASFT-XL (ours) | 1.28M | 733M | 870M | 100 | 3.74 | 228.8 |

Table 7: Comparison of ASFT and baselines in # params and training cost (*i.e.* product of batch size and training iterations). Some numbers are borrowed from Crowson et al. (2024).

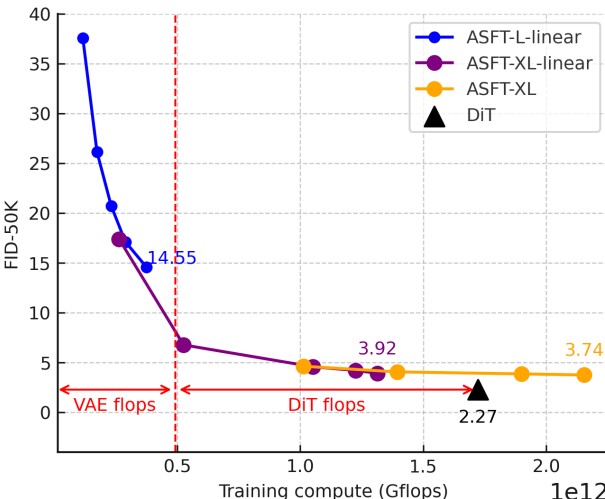

Figure 5: Comparing the performance vs total training compute comparison of ASFT and DiT (Peebles & Xie, 2023).

## B  PERFORMANCE VS TRAINING COMPUTE

We compare the performance vs total training compute of ASFT and DiT (Peebles & Xie, 2023) in Gflops. ASFT-linear denotes the variant of ASFT where the cross-attention in the spatial aware encoder is replaced with grouping followed by a linear layer. We found this could be an efficient variant of standard ASFT while still achieving competitive performance. Fig. 5 shows the comparison of the training compute in Gflops vs FID-50K between ASFT and latent diffusion model DiT (Peebles & Xie, 2023) including the tranining compute of the first stage VAE. We estimate the training cost of VAE based the model card listed in HuggingFace[2]. As shown, the training cost of VAE is not negligible and reasonable models with FID ≈ 6.5 can be trained for the same cost.

Admittedly, under equivalent training Gflops, ASFT achieves comparable but not as good performance as DiT in terms of FID score (with a difference smaller than 1.65 FID points). We attribute this gap to the fact that DiT's VAE was trained on a dataset much larger than ImageNet, using a domain-specific architecture (*e.g.* a convolutional U-Net). We believe that the simplicity of implementing and training ASFT models in practice, and the trivial extension to different data domains (as shown in Sect. 4.3) are strong arguments to counter an FID difference of smaller than 1.65 points.

---

[2]https://huggingface.co/stabilityai/sd-vae-ft-mse

## C  ARCHITECTURE ABLATION

We also provide an architecture ablation in Tab. 8 showcasing different design decisions. We compare two variants of Transformer-based architectures ASFT: a vanilla PerceiverIO that directly operates on ambient space, but without using spatial aware latents and ASFT. As it can be seen, the spatially aware latents introduced in ASFT greatly improve performance across all metrics in the image domain, justifying our design decisions. We note that we did not observe the same large benefits for 3D point clouds, which we hypothesize can be due to their irregular structure.

| Model | FID($\downarrow$) | Precision($\uparrow$) | Recall($\uparrow$) |
|---|---|---|---|
| PerceiverIO | 65.09 | 0.38 | 0.01 |
| ASFT (ours) | **7.03** | **0.69** | **0.34** |

Table 8: Benchmarking vanilla PerceiverIO and ASFT with spatially aware latents on LSUN-Church-256 (Yu et al., 2015).

## D  RESOLUTION AGNOSTIC GENERATION: QUANTITATIVE ANALYSIS

Due to the fact that ASFT decodes each coordinate-value pair independently given $z_{f_t}$, during inference time one can decode as many coordinate-value pairs as desired, therefore allowing resolution to change during inference. We now quantitatively evaluate the performance of ASFT in this setting. In Tab. 9 we compare the FID of different recipes. First, ASFT is trained on FFHQ-256 and bilinear or bicubic interpolation is applied to generated samples to get images at 512. On the other hand, ASFT can directly generate images at resolution 512 by simply increasing the number of coordinate-value pairs during inference without further tuning. As shown in Tab. 9 , ASFT achieves lower FID when compared with other manually designed interpolation methods, showcasing the benefit of developing generative models on ambient space.

| | ASFT | Bilinear | Bicubic |
|---|---|---|---|
| FID($\downarrow$) | 23.09 | 35.05 | 24.34 |

Table 9: FID of different super resolution methods to generate images at resolution $512 \times 512$ for ASFT trained on FFHQ-256.

## E  ADDITIONAL IMAGENET SAMPLES

We show uncurated samples of different classes from ASFT-XL trained on ImageNet-256 in Fig. 6 and Fig. 7. Guidance scales in CFG are set as $4.0$ for loggerhead turtle, macaw, otter, coral reef and $2.0$ otherwise.

## F  ADDITIONAL SHAPENET SAMPLES

We show uncurated samples from ASFT-L trained jointly on 55 ShapeNet categories in Fig. 8 and Fig. 9.

## G  IMAGE TO POINT CLOUD GENERATION ON OBJAVERSE

We also showcase that ASFT can directly integrate conditional information like images. We train an image-to-point-cloud generation model on Objaverse (Deitke et al., 2023), which contains 800k 3D objects of wide variety, to illustrate the capability of ASFT on larger-scaled 3D generative tasks. In particular, conditional information (i.e., an image) is integrated to our model through cross-attention. For each object in Objaverse, we sample point cloud with 16k points. To get images for conditioning, each object is rendered with 40 degrees field of view, $448 \times 448$ resolution, at 3.5 units on the opposite

| Model | ULIP-I $\uparrow$ | P-FID $\downarrow$ |
|---|---|---|
| Shap-E (Jun & Nichol, 2023) | 0.1307 | - |
| Michelangelo (Zhao et al., 2024) | 0.1899 | - |
| CLAY (Zhang et al., 2024) | 0.2066 | 0.9946 |
| ASFT (ours) | **0.2976** | **0.3638** |

Table 10: Image-conditioned 3D point cloud generation performance on Objaverse.

sides of x and z axes looking at the origin. We extract features via DINOv2 (Oquab et al., 2023) which is concatenated with Plucker ray embedding (Plucker, 2018) of each patch in DINOv2 feature. In each block, the learnable latent vector $z_{f_t}$ cross attends to image feature. During training, the image conditioning is dropped randomly with 10% probability. Therefore, our model can also benefit from popular classifier-free guidance (CFG) to increase the guidance strength. The model is trained with batch size 384 for 500k iterations. During sampling, we use an Euler-Maruyama sampler (Ma et al., 2024) with 500 steps to generate point clouds.

Tab. 10 lists the performance of ASFT in comparison of recent baselines on Objaverse. We report ULIP-I (Xue et al., 2024) and P-FID (Nichol et al., 2022) following CLAY (Zhang et al., 2024). PointNet++ (Qi et al., 2017a;b; Nichol et al., 2022) is employed to evaluate P-FID. ULIP-I is an analogy to CLIP for text-to-image generation. ULIP-I is measured as the cosine similarity between point-cloud features from ULIP-2 model (Xue et al., 2024) and image features from CLIP model (Radford et al., 2021). Numbers of baseline models are directly borrowed from CLAY (Zhang et al., 2024). We calculate the metrics of our ASFT on 10k sampled point clouds. In our case, P-FID is measured on point clouds with 4096 points following Shape-E (Jun & Nichol, 2023) while ULIP-I is measured on point clouds with 10k points following ULIP-2 (Xue et al., 2024). Note that since CLAY (Zhang et al., 2024) is not open-source, we do not have the access to the exact evaluation setting or the conditional images rendered from Objaverse. But all evaluation settings of ASFT are provided for reproduction purpose. As shown in Tab. 10, our ASFT achieves strong performance on large-scaled image-conditioned 3D generative tasks. Compared with CLAY (Zhang et al., 2024), which is a 2-stage latent diffusion model, ASFT demonstrates very strong performance on both ULIP-I and P-FID.

Fig. 10 show examples of sampled point clouds and corresponding conditional images. As discussed in §4.4, ASFT on Objaverse also enjoys the flexibility of resolution agnostic generation. The right columns in Fig. 10 show results sampled with more points than what the model is trained on. As shown, ASFT learns to generate 3D objects with rich details that match the conditional images ultimately being able to generate a continuous surface.

## H ADDITIONAL RESOLUTION AGNOSTIC IMAGE SAMPLES

We show additional samples generated at different resolutions from ASFT trained on ImageNet-256 in Fig. 11.

972
973
974
975
976
977
978
979
980
981
982
983
984
985
986
987
988
989
990
991
992
993
994
995
996
997
998
999
1000
1001
1002
1003
1004
1005
1006
1007
1008
1009
1010
1011
1012
1013
1014
1015
1016
1017
1018
1019
1020
1021
1022
1023
1024
1025

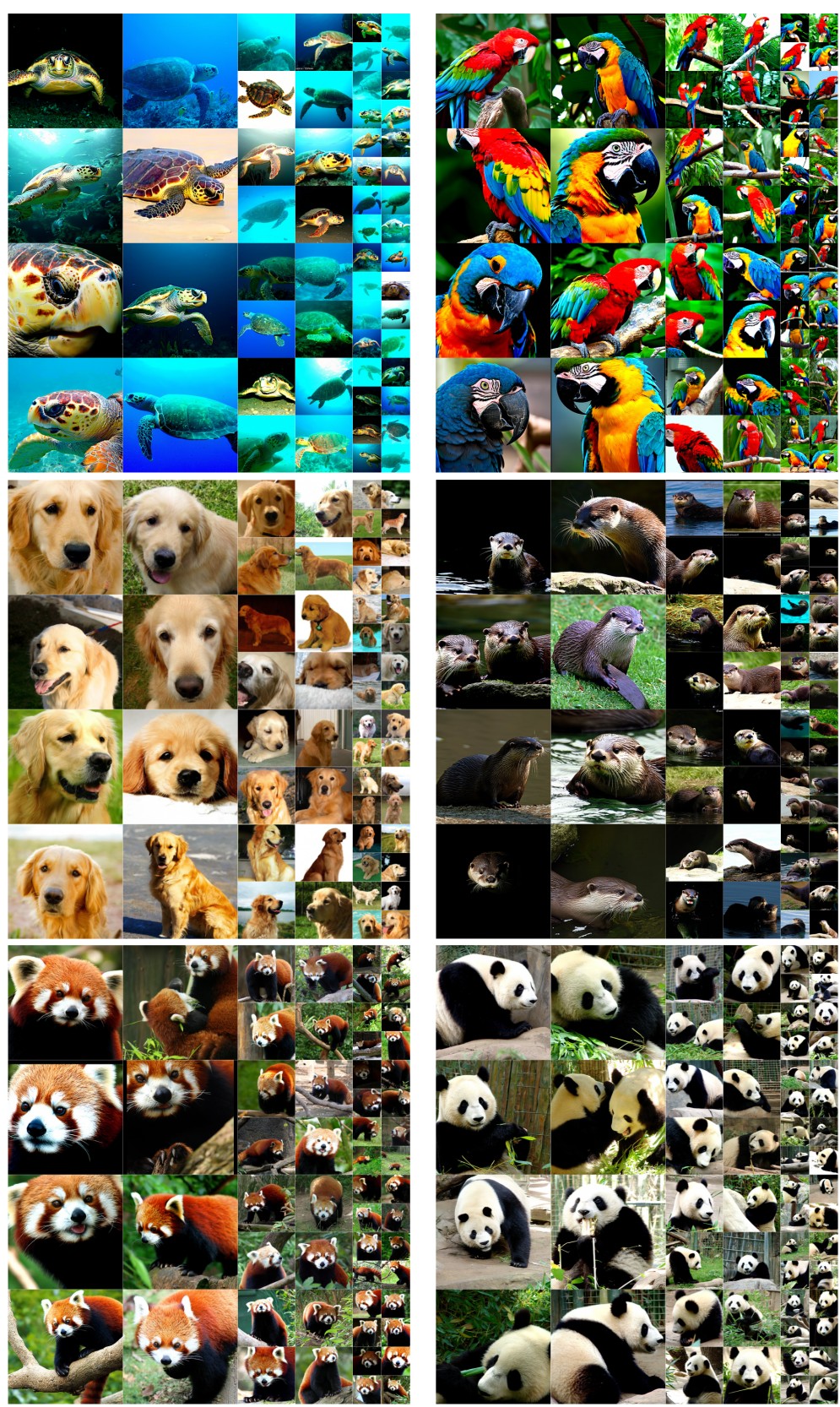

Figure 6: Uncurated samples of class labels: loggerhead turtle (33), macaw (88), golden retriever (207), otter (360) and red panda (387), and panda (388) from ASFT trained on ImageNet-256.

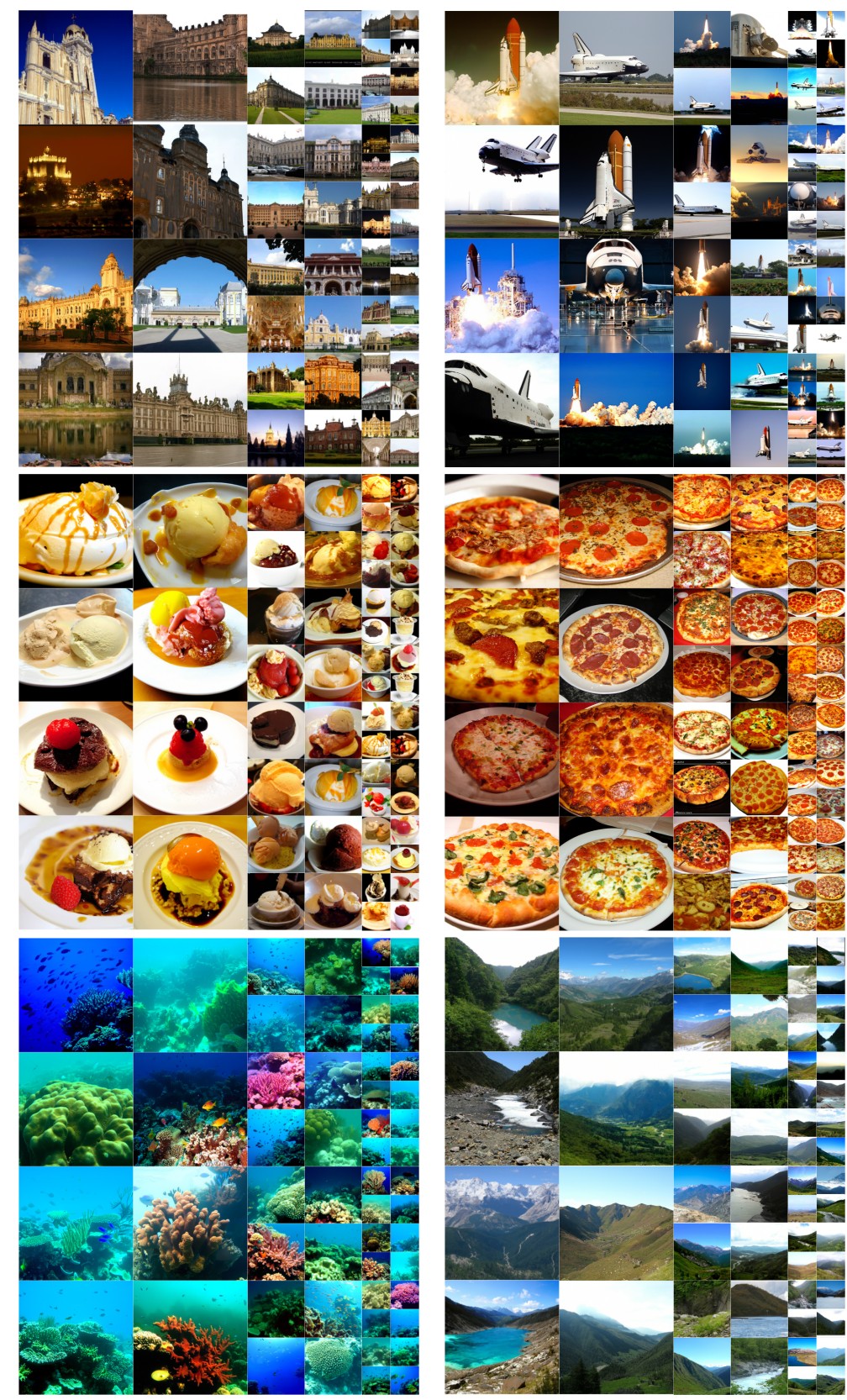

Figure 7: Uncurated samples of class labels: palace (698), space shuttle (812), ice cream (928), pizza (963), coral reef (973), and valley (979) from ASFT trained on ImageNet-256.

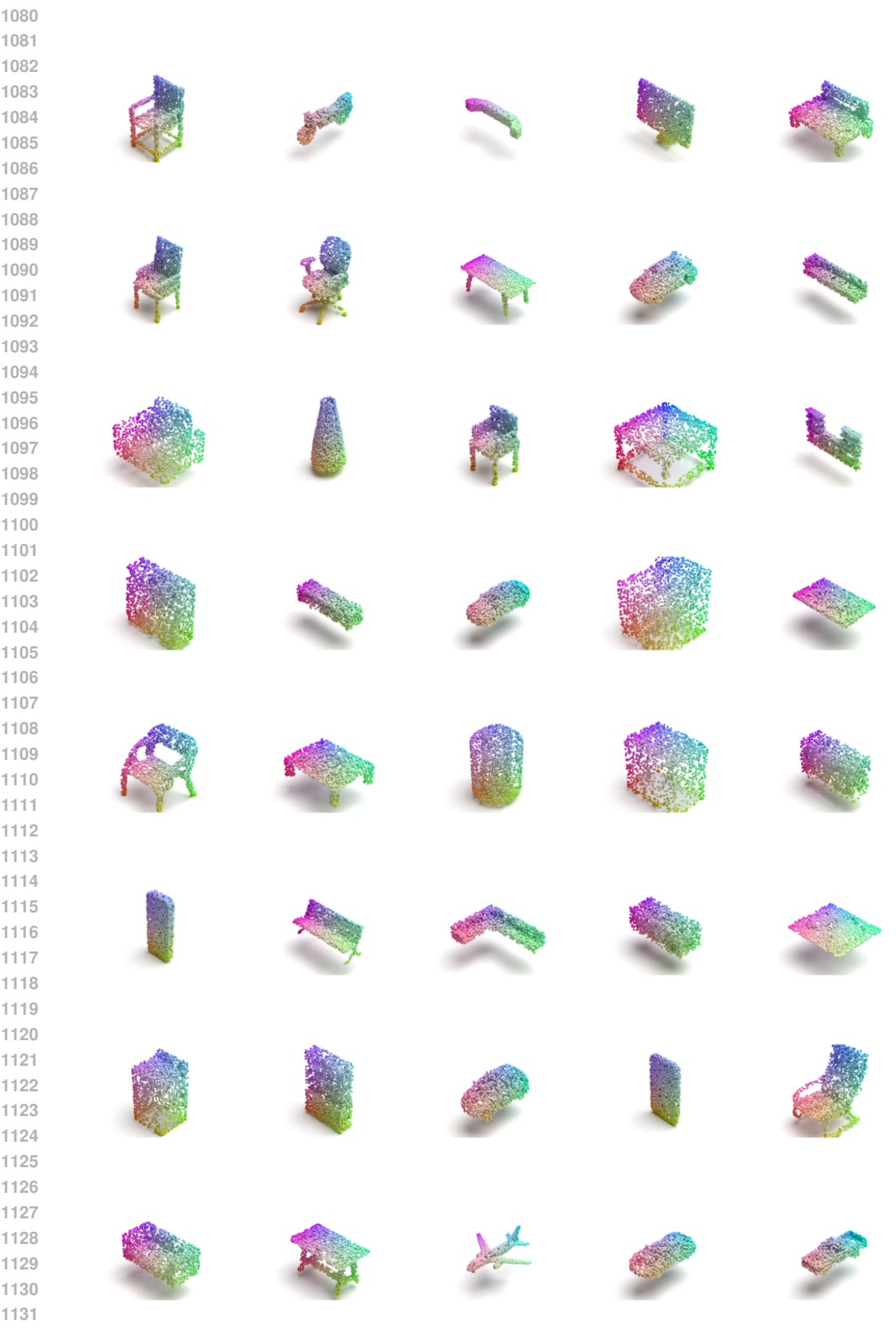

Figure 8: Additional uncurated ShapeNet generations using 2048 points from the unconditional model jointly trained on 55 categories

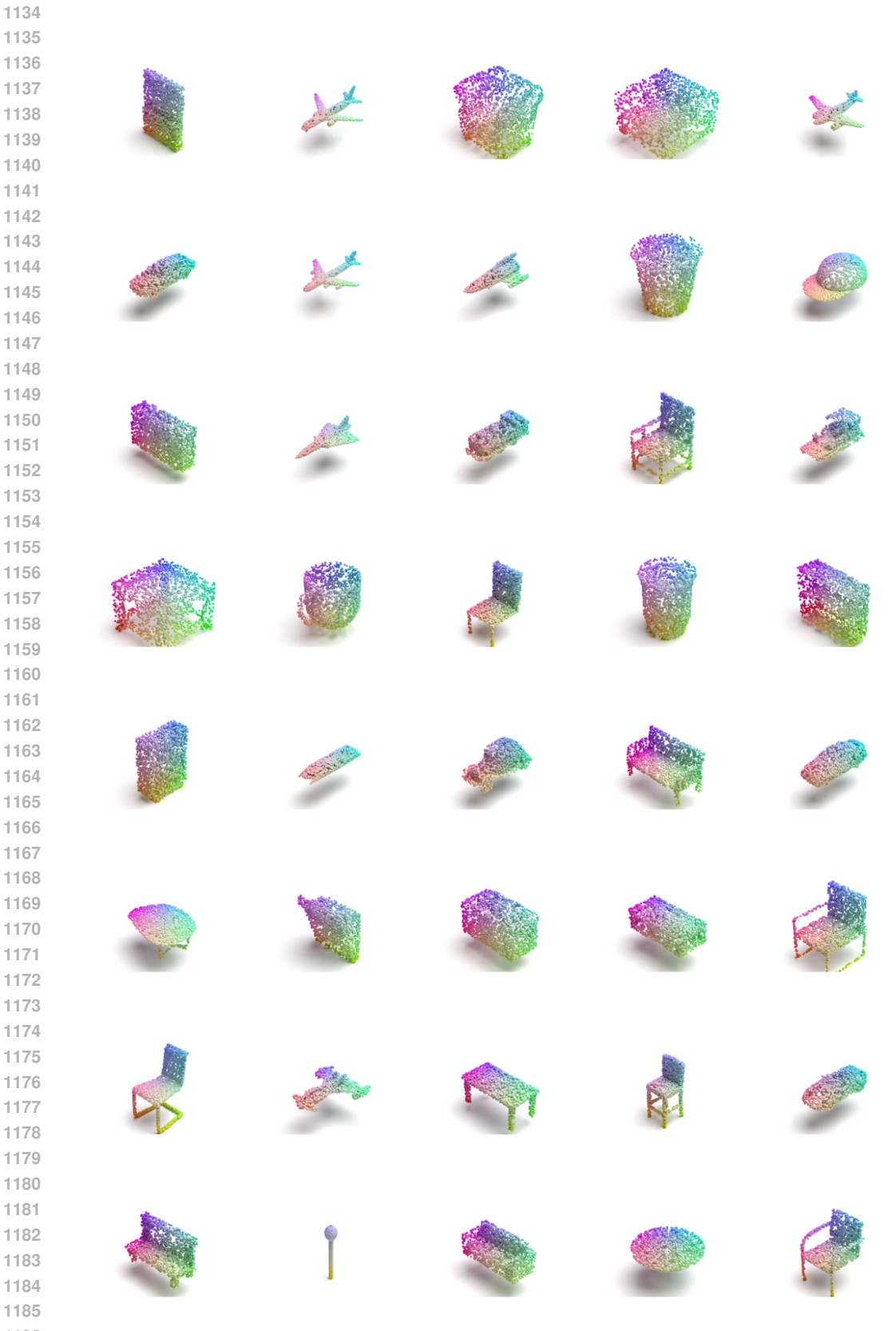

Figure 9: Additional uncurated ShapeNet generations using 2048 points from the unconditional model jointly trained on 55 categories

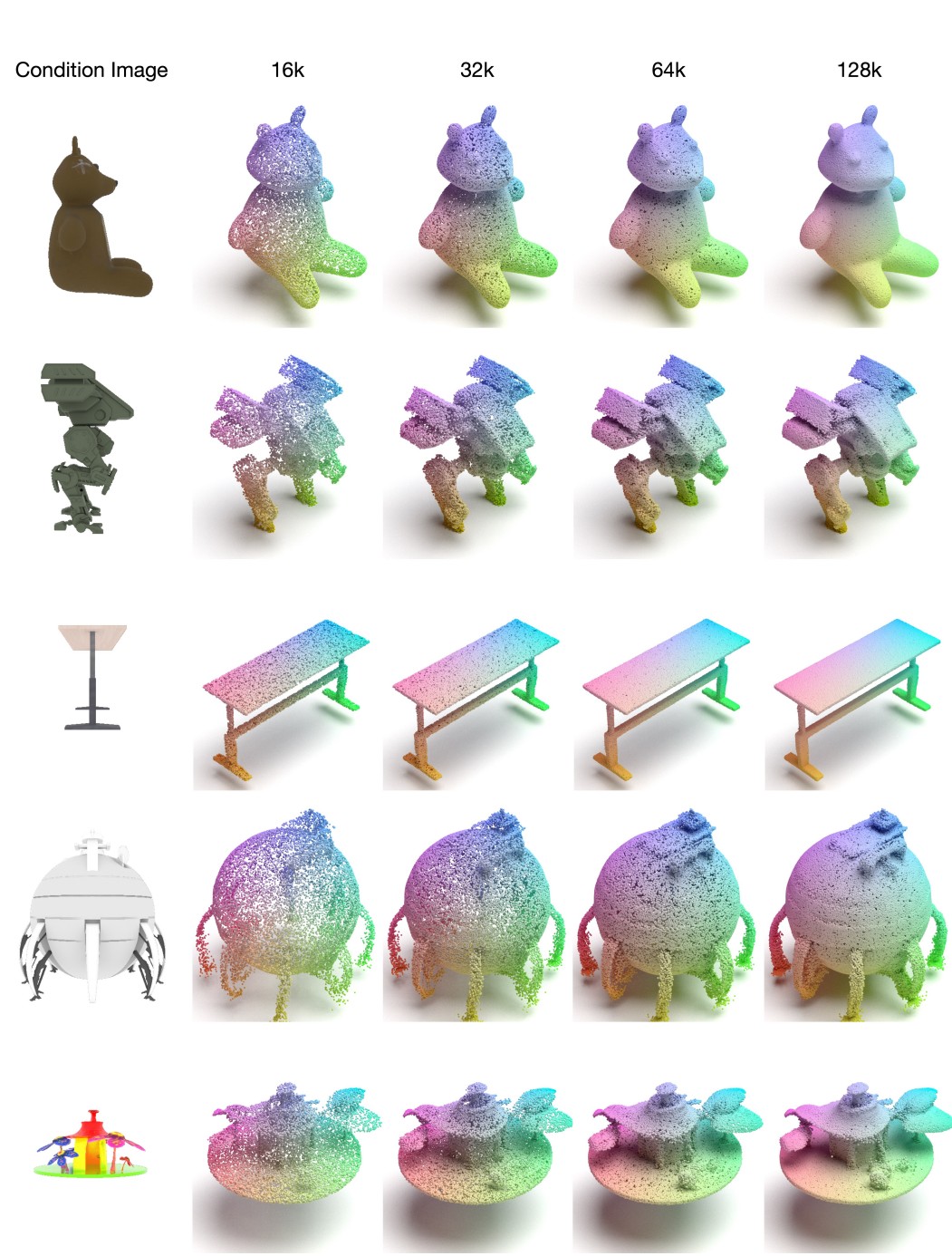

Figure 10: Image-conditioned point clouds with 16k, 32k, 64k, and 128k points generated from an ASFT trained on Objaverse (training with 16k points, CFG scale 5.0).

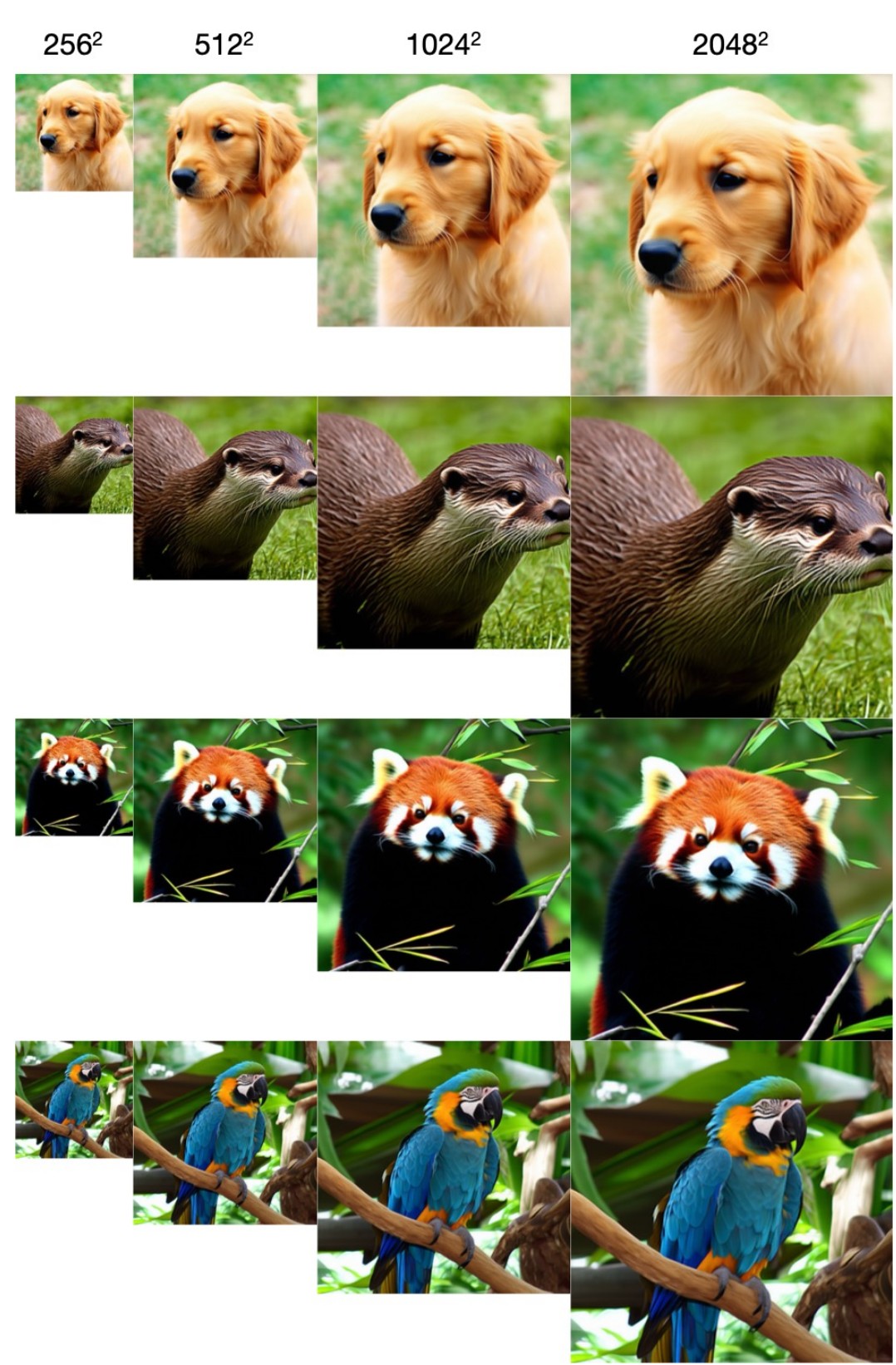

Figure 11: Images generated at 256, 512, 1024, and 2048 resolutions from an ASFT trained on ImageNet-256.

