# OpenReview forum: "Coordinate In and Value Out: Training Flow Transformers in Ambient Space"
_ICLR.cc/2025/Conference — Submitted to ICLR 2025_

### Official Review · Reviewer_pE6r · 2024-11-03

**Soundness:** 2
**Presentation:** 2
**Contribution:** 2
**Rating:** 3
**Confidence:** 4

**Summary:**

This paper claims that building latent-space diffusion models have several shortcomings such as non-end-to-end optimization and the requirement of domain-specific compression (e.g., VAE) models, and thus designs an ambient-space flow transformer (ASFT) architecture for generative tasks.

**Strengths:**

1. The analyses of the potential shortcomings of existing latent diffusion models are presented. Based on the motivations, this paper constructs a single-stage learning model,
2. Learning over coordinate-value pairs facilitates data-agostic processing. In this way, an image is actually treated as a "colored point cloud".

**Weaknesses:**

1. Although the authors analyzed some aspects of potential drawbacks of latent-space diffusion models, it is hard to be convinced that ambient-space (e.g., pixel-space, point-space) learning is a more promising direction:
-- First, I don't think training domain-specific data compressors is a cumbersome practice that should be criticized. When some powerful VAEs are already trained and released, the community can directly use them. There aren't that many types of data modalities. Building compressors for each type of data (e.g., image, video, point cloud, mesh) is totally acceptable.
-- Second, the current mainstream practice is to separately train the data compressor and the subsequent latent diffusion model, but it does not mean that such training workflow cannot be made end-to-end. We can't say it is a drawback just because we haven't explored it. Generative models evolve so fast. I think making it end-to-end is not impossible.
-- Third, anyway, for now the great success of various latent diffusion models seems to demonstrate the superiority of learning in the latent space instead of the ambient space.

2. Building generative models with coordinate-value pairs may essentially restrict its application scenarios and conditional generation capabilities. For 3D generation, point cloud is apparently not the final choice. What we want is the continuous surface, together with textures. Existing 3D generative models either use meshes or implicit fields. However, the proposed method faces difficulties in generating such data. Besides, I notice that the proposed method is only implemented with class label conditioned generation, which is quite out-of-date. Its conditional generation capabilities (e.g., text-guided, image-guided) are questionable.

3. The experimental settings are not very persuasive. For image generation, the model is trained on ImageNet. For point cloud generation, the model is trained on ShapeNet. The experimental results cannot demonstrate the potential of the proposed method for learning from larger-scale high-quality data, such as LAION and Objaverse datasets. It can be observed that the diversity and quality of the generated images obviously cannot catch up the current state-of-the-art latent image diffusion models. The generated point clouds are also noisy and lack details, especially when compared with state-of-the-art 3D native diffusion models (like CLAY).

**Questions:**

1. Does the proposed generation framework support more practical conditioning mechanism (e.g., guided by text or single image)?
2. Can the proposed learning model be scaled up using larger-scale high-quality datasets such as LAION and Objaverse?
3. The authors are suggested to provide further explanations about how the latent variable z_{f_t} for context encoding is obtained.

---

> ### Author Response · Authors · 2024-11-20
> **Official Response to Reviewer pE6r**
>
> 1. Question about whether ambient-space (e.g., pixel-space, point-space) learning is a promising direction.
>     - Admittedly, latent generative modeling with two-stage training paradigm has been gaining popularity recently. However, we want to point out that this does not diminish the value of exploring generative models in ambient space. As a matter of fact, ambient/pixel space models do surpass latent space models on ImageNet-256, Tab. 3 (eg. VDM++ U-ViT vs DiT), **there’s no empirical reason to believe in the superiority of learning in latent space instead of pixel space.**
>     - We kindly disagree that building domain-specific data compressors for each data type is a non issue. To design different VAEs for different data domains, efforts are required to curate architectures, training recipes and datasets. The efforts put on building a compressor for one data domain do not usually transfer to other domains.
>     - Finally, as the reviewer pointed out “[... Generative models evolve so fast. I think making it end-to-end is not impossible ...]” Our contribution in this submission is exactly exploring an approach to build an end-to-end generative framework in a data-agnostic way. We aim to build a simple and flexible generative model that allows to tackle multiple data domains in a single training stage.
>
> 2. Building generative models with coordinate-value pairs may essentially restrict its application scenarios and conditional generation capabilities.
>     - We want to kindly point out that the coordinate-value pairs are not constraint for broader applications. On the contrary, modeling data as coordinate-value pairs actually enables ASFT to be directly applied in different data domains. ASFT actually models an explicit neural field based on coordinate-value pairs. Since all outputs are predicted conditionally independent with each other given the learnable latent $z_{f_t}$. This means that given the latent $z_{f_t}$ we can continuously query the representation like implicit fields do. As an illustrative example, we can generate infinitely dense pointclouds that represent the continuous surface in 3D space (see Fig. 4 in the paper where we sample pointclouds with 100K points, also see Fig. 10 where sample pointclouds with 128k points from models trained on Objaverse). **ASFT does not suffer from generating implicit fields, as a matter of fact, it’s designed to predict explicit neural fields that can be continously evaluated. **
>     - We benchmark sparse 3D point cloud generation mainly because (1) it’s unstructured data unlike images which provide a testbed for more diverse data domains and (2) point cloud generation has well-established benchmarks to compare our results with existing work.
>
> 3. Besides, the proposed method is only implemented with class label conditioned generation, which is quite out-of-date. Its conditional generation capabilities (e.g., text-guided, image-guided) are questionable.
>     - **We note that class-conditioned image generation on ImageNet is a widely used benchmark for generative modeling of images and it is considered the standard benchmark for evaluation. Therefore, we include benchmark our performance on ImageNet128 and ImageNet256.** ASFT can directly integrate other conditional information (as latent diffusion models do). In light of your valuable suggestion, we are training an image-to-point-cloud generation model on Objaverse to showcase the conditional generation capabilities. In particular,  the conditioning (i.e., image) is integrated into ASFT through cross-attention. In each block, the latent vector $z_{f_t}$ cross attends to image features from DINOv2. During training, the image conditioning is dropped randomly with 10% probability. Therefore, our model can also benefit from popular classifier-free guidance (CFG) to enhance the match between samples and conditions.

---

> > ### Author Response · Authors · 2024-11-20
> > **Official Response to Reviewer pE6r (2)**
> >
> > 4. Can the proposed learning model be scaled up using larger-scale high-quality datasets such as LAION and Objaverse?
> >     - In light of the valuable suggestion, we have added results on point cloud generation of Objaverse. We train an image-to-point-cloud ASFT model on Objaverse, containing more than 800k 3D objects of wide variety. Compared with SOTA 3D generative models like CLAY, our ASFT demonstrates competitive performance on image-to-point-cloud generation. These results further demonstrate the generalizability of proposed ASFT on diverse image and point cloud generative tasks. More details regarding Objaverse experiments can be found in Appendix G.
> >         | Model | ULIP-I (↑) | P-FID (↓) |
> >         | -------- | ------- | -------- |
> >         | Shap-E           | 0.1307 | - |
> >         | Michelangelo | 0.1899 | - |
> >         | CLAY              | 0.2066 | 0.9946 |
> >         | ASFT (ours)   | **0.2976** | **0.3638** |
> >
> > 5. The authors are suggested to provide further explanations about how the latent variable z_{f_t} for context encoding is obtained.
> >     - Thanks for your suggestion, we have updated the text in the paper to reflect this (changes are highlighted in red): “Latent vectors $z_{f_t} \in \mathbb{R}^{L \times D}$ are learned in an end-to-end manner. In particular, the learnable latent $z_{f_t}$ cross-attends to input coordinate-value pairs of noisy data at a given timestep $t$. Latent vectors are spatial-aware, this means that each of the $L$ latents only attends to a set of neighboring coordinate-value pairs. Latent vectors are then updated using several self-attention blocks.”
> >
> > References:
> >
> > [1]  A. A. Elhag et. al, '“Manifold Diffusion Fields”, International Conference on Learning Representations 2024, https://openreview.net/forum?id=BZtEthuXRF

---

### Official Review · Reviewer_N7EZ · 2024-11-03

**Soundness:** 3
**Presentation:** 3
**Contribution:** 3
**Rating:** 6
**Confidence:** 2

**Summary:**

The paper introduces **Ambient Space Flow Transformers (ASFT)** as a novel approach to generative modeling that simplifies the training process by eliminating the need for latent space data compressors. ASFT works directly in the ambient space (i.e., the original data domain), aiming to be a domain-agnostic model applicable across different types of data, such as images and 3D point clouds.

**Strengths:**

- **Simplicity and Efficiency**: ASFT's single-stage training process is simpler than traditional two-stage models, making it easier to implement and tune.
- **Versatility**: It is designed to work across multiple data types without modifications, unlike models that require domain-specific architectures.
- **Resolution Scalability**: The ability to generate high-resolution outputs from lower-resolution training data provides flexibility and potential computational savings.
- **Competitive Results**: Despite its simplicity, ASFT achieves strong performance metrics comparable to state-of-the-art models across images and point clouds.

**Weaknesses:**

- **Potential for Lower Fidelity in Complex Domains**: ASFT may not match latent space models in specific metrics, such as Fréchet Inception Distance (FID), particularly when latent space models are pre-trained on extensive datasets.
- **Dependence on Model Size for Best Results**: The model's performance improves significantly with scale, which may lead to high computational costs for larger ASFT versions, particularly in comparison with models leveraging pre-trained compressors.
- **Challenges with High Dimensional Data**: For very high-resolution applications, ASFT’s point-wise approach might face optimization difficulties due to the increased complexity in decoding large numbers of coordinate-value pairs.

**Questions:**

Despite it is a domain-agnostic model, are there any possible way to include domain-specific knowledges to further boost the quality?

---

> ### Author Response · Authors · 2024-11-20
> **Official Response to Reviewer N7EZ**
>
> 1. Q: Potential for Lower Fidelity in Complex Domains
>     - We want to note that when comparing to baselines trained in ambient space like Simple Diffusion (U-Net), RIN, HDiT, our proposed ASFT achieves comparable performance despite being domain-agnostic and readily applied to other domains like 3D point cloud generation (see our new results on Objaverse in Sect G of Appendix). Other baselines with large model sizes containing 2B parameters (Simple Diff U-ViT and VDM++ U-ViT) achieve better performance than ASFT, which we attribute to their model size being 3 times bigger than ASFT-XL.
>     - Admittedly, there is a performance gap compared with models using latent space with pre-trained VAEs. However, we want to point out that these VAEs are trained on a much larger dataset than ImageNet whereas ASFT is only trained on ImageNet. In particular, the widely used SD-VAE (https://huggingface.co/stabilityai/sd-vae-ft-mse) used for latent space generative modeling, is first trained on OpenImages (containing ~9M images) and then finetuned on subset of LAION (containing over 238k images) for a total of ~9.23M images. Whereas ImageNet contains ~1.28M images. **This means that ASFT is trained on 13% of the data used to train DiT and SiT.** We have updated Tab. 3 and Tab. 7 in the appendix to reflect this.
>
> 2. Q: Dependence on Model Size for Best Results
>     - We agree with reviewer that the model benefits from increasing model size. However, we want to highlight that this actually shows that ASFT benefits from scale as widely studied in many diffusion models. Latent diffusion models also rely on scaling up the model size to achieve better performance. In fact, our largest model ASFT-XL has comparable model size as DiT-XL and SiT-XL.  Investigating efficient variants of current version can be a valuable direction and we’ll include this discussion in future works of updated manuscript.
>
> 3. Q: Challenges with High Dimensional Data
>     - For high dimensional data, we can subsample the decoded coordinate-value pairs as shown in Figure 3b during training to decrease the FLOPs. As a future direction, more efficient architectures like SSM can be applied. In addition, note that because ASFT models coordinate-value maps it can generate samples in arbitrary dimensions. We have also included additional examples of samples of higher resolution (i.e., 2048$\times$2048) from ASFT trained on ImageNet-256 in Appendix H. It indicates that ASFT can handle high-resolution data in inference trivially.
>
> 4. Q: Despite it is a domain-agnostic model, are there any possible way to include domain-specific knowledges to further boost the quality?
>     - We agree with the reviewers that integrating domain-specific knowledge can boost the performance of proposed ASFT. For instance, one might take into account the power law of the frequency spectrum of images when designing coordinate-value maps [1] or add regularization between our ASFT and other representation learning models [2]. However, we want to kindly highlight that the scope of our work is to demonstrate a way of building unified generative models that can be applied to different domains with little to no domain-specific tweaking. As shown in the paper, this domain-agnostic framework already achieves comparable performance on image and point cloud generation as curated models for each domain. We believe this indicates a promising direction of building unified and flexible generative model.
>
> References:
>
> [1]  Rissanen, Severi, et. al. "Generative modelling with inverse heat dissipation." arXiv preprint arXiv:2206.13397 (2022).
>
> [2] Yu, Sihyun, et al. "Representation alignment for generation: Training diffusion transformers is easier than you think." arXiv preprint arXiv:2410.06940 (2024).

---

### Official Review · Reviewer_TVCm · 2024-11-04

**Soundness:** 4
**Presentation:** 4
**Contribution:** 3
**Rating:** 6
**Confidence:** 4

**Summary:**

The paper presents Ambient Space Flow Transformer (ASFT), a flow matching generative model working with an implicit representation of the data (INR), instead of a latent one. A modified version of PerceiverIO used as a backbone allows both images and 3D point clouds to bo modeled without any design changes.

**Strengths:**

1. The novelty is good - single stage generation via flow matching on function space without any pretrained encoder-decoder models.
2. On images, ASFT beats function space approaches on FID.
3. On 3D point clouds, the method is better or on par with other models.
4. Due to the independence of the input coordinates, the model allows sub-sampling of point clouds and super-resolution of images.
5. The writing is very good. The information flow is maintained and all of the ideas are clearly explained.

**Weaknesses:**

1. The image results presented are in low resolution only - 128x128 and 256x256.
2. The overall image results are not very convincing - 128x128 images are not a solid proof for superiority of the model. Meanwhile, results on 256x256 are worse than many baselines (even though domain-specific and using pretrained models).

**Questions:**

1. Why is the unified representation important? From the practical point of view, it doesn't make that much of a difference to have two domain-specific backbones instead of a shared one (at least for images and point clouds). If we sacrifice the unification, is there any better backbone choice, especially for images that would result in better results and/or scalability?
2. The model is shown only for images and 3D point clouds. What about other modalities? Are there any additional challenges?
3. Does the image model scale to higher resolutions?
4. Figure 2 may suggest both image and 3D coordinates are passed to the network at the same time. The authors should consider clearly separating them in the figure and updating the caption to highlight this.
5. In Figure 3 (b) color palette is hard to read - the differences should be more visible.
6. It would be interesting to see a comparison with standard image upscaling in Figure 4 (a).
7. In introduction, citations for VAE, VQVAE, VQGAN, transformers, PointNet, UNet are missing.
8. "UNet" and "U-Net" used - please pick one.

---

> ### Author Response · Authors · 2024-11-20
> **Official Response to Reviewer TVCm**
>
> 1. Q: The overall image results are not very convincing - 128x128 images are not a solid proof for superiority of the model. Meanwhile, results on 256x256 are worse than many baselines (even if they are pretrained and domain-specific).
>     - We want to highlight that comparing to baselines that are also trained on ambient space like Simple Diffusion (U-Net), RIN and HDiT, we acheive comparable performance while formulating a domain-agnostic approach that is directly applicable to other domains like 3D generation. Other baselines with large model sizes containing 2B parameters (Simple Diff U-ViT and VDM++ U-ViT) achieve better performance than ASFT, which we attribute to their model size being 3 times bigger than ASFT-XL, see Tab. 7 in the Appendix. We have updated Tab. 3 to also reflect this.
>     - There is a performance gap compared with models using latent space with pre-trained VAEs. However, we want to point out that these VAEs are trained on a much larger dataset than ImageNet whereas ASFT is only trained on ImageNet. In particular, the widely used SD-VAE (https://huggingface.co/stabilityai/sd-vae-ft-mse) used for latent space generative modeling, is first trained on OpenImages (containing ~9M images) and then finetuned on subset of LAION (containing over 238k images) for a total of ~9.23M images. Whereas ImageNet contains ~1.28M images. **This means that ASFT is trained on 13% of the data used to train DiT and SiT.** We have updated Tab. 3 and Tab. 7 in the appendix to reflect this.
>
> 2. Q: Why is the unified representation important? From the practical point of view, it doesn't make that much of a difference to have two domain-specific backbones instead of a shared one (at least for images and point clouds). If we sacrifice the unification, is there any better backbone choice, especially for images that would result in better results and/or scalability?
>     - We thank the reviewer for bringing up this point. Our hypothesis is that the ultimate goal of generative modeling is to train models on every existing bit of information. Unfortunately, these information bits are distributed across different data domains (ie. there are bits of information in images that are not captured by text datasets). A unified generative framework that can leverage different data modalities seamlessly is therefore an important direction to pursue.
>     - As the reviewer points out, theres a tradeoff to explore when designing generative models,  we can sacrifice the unification (ie. and its simplicity across domains and ability to scale up the training data) to benefit from more efficient domain-specific approaches. In particular, one may benefit from optimizing architectures for each modality separately. For example, we could make use of domain-specific biases like the frequency spectrum of images following a power law [4] or the fact that 3D models are multi-view consistent [5]. However, we want to point out that a trend recent works demonstrate, a Transformer-based architectures that trivially benefits from scaling has achieved superior performance in many applications like image [6], video [6], or even graph structured data [7]. We believe ASFT represents a promising in this direction, where we leverage  Transformer-based domain-agnostic generative models that can be effectively trained.
>
> 3. Q: The model is shown only for images and 3D point clouds. What about other modalities? Are there any additional challenges?
>     - Our model can be applied to other modalities with ease. Once certain modality is formulated as a mapping from coordinate space to signal space, ASFT can be directly applied to this data modality. For example, to build generative model on non-Euclidean spaces (eg. graphs or Riemannian manifolds), we can use an intrinsic coordinate system based on eigen-decomposition to define coordinate-value maps, as suggested in [2]. We specifically opted for images and point clouds since images are structured and dense in 2D space while point clouds are sparse and unstructured representations in 3D. These two settings cover most of the use cases for other data domains.

---

> > ### Author Response · Authors · 2024-11-20
> > **Official Response to Reviewer TVCm (2)**
> >
> > 4. Q: Does the image model scale to higher resolutions?
> >     - As shown in Figure 4, ASFT allows sampling in a resolution-free manner. Namely, it allows sampling at higher resolution than it was trained on. Appendix H in updated manuscript also showcases that ASFT can trivially generate images of high resolution at 2048 in inference. ASFT can also be trained at higher resolutions, which typically require more training FLOPs. In this setting, one can employ efficient architectures through strategies like token merging [1] or masking [3]. We believe this could a valuable direction to explore in future work.
> >
> > 5. Q: Figure 2 may suggest both image and 3D coordinates are passed to the network at the same time. The authors should consider clearly separating them in the figure and updating the caption to highlight this.
> >     - We have updated Figure 2 to better illustrate the pipeline. We’ll also add clarification in the caption in the updated manuscript.
> >
> > 6. Q: In Figure 3 (b) color palette is hard to read - the differences should be more visible.
> >     - We have updated Fig. 3b to make the color more readable.
> >
> > 7. Q: It would be interesting to see a comparison with standard image upscaling in Figure 4 (a).
> >     - We want to kindly point out that we compare ASFT and standard upsampling strategies like bilinear and bicubic interpolation in Tab. 9. As shown, given a ASFT trained at dataset with resolution 256, directly sampling with resolution 512 achieves better performance than standard interpolation methods. It indicates the benefit of developing generative models on ambient space like ASFT.
> >
> > 8. Q: In introduction, citations for VAE, VQVAE, VQGAN, transformers, PointNet, U-Net are missing.
> >     - We have added the citations to the papers in the updated manuscript.
> >
> > 9. Q: "UNet" and "U-Net" used - please pick one.
> >     - We have fixed the spelling in the updated manuscript.
> >
> > References:
> >
> > [1] Bolya, Daniel et. al. "Token merging: Your vit but faster." arXiv preprint arXiv:2210.09461 (2022).
> >
> > [2] A. A. Elhag et. al, '“Manifold Diffusion Fields”, International Conference on Learning Representations 2024, https://openreview.net/forum?id=BZtEthuXRF
> >
> > [3] Sehwag, Vikash, et al. "Stretching Each Dollar: Diffusion Training from Scratch on a Micro-Budget." arXiv preprint arXiv:2407.15811 (2024).
> >
> > [4] Rissanen, Severi, et. al. "Generative modelling with inverse heat dissipation." arXiv preprint arXiv:2206.13397 (2022).
> >
> > [5] Shi, Yichun, et al. "Mvdream: Multi-view diffusion for 3d generation." arXiv preprint arXiv:2308.16512 (2023).
> >
> > [6] Jabri, Allan et al. "Scalable adaptive computation for iterative generation." arXiv preprint arXiv:2212.11972 (2022).
> >
> > [7] Wang, Yuyang, et al. "Swallowing the Bitter Pill: Simplified Scalable Conformer Generation." Forty-first International Conference on Machine Learning.

---

> > > ### Comment · Reviewer_TVCm · 2024-11-25
> > >
> > > I thank the authors for addressing my questions.
> > >
> > > The idea of unifying all data modalities into a single generative model is an interesting future direction, and the proposed model is a good contribution towards this goal. The current results, however, are promising but still not very convincing. The next step should be an attempt to scale the model and/or training data size for a better comparison with other methods. It would be also good to see other modalities included.
> > >
> > > After leveraging the pros and cons, and reading other reviews, I maintain my score.

---

> > > > ### Author Response · Authors · 2024-11-25
> > > > **Official Response to Reviewer TVCm (3)**
> > > >
> > > > We thank the reviewer for the timely reply and valuing our contributions. We agree scaling the current framework is a promising direction as the experimental results on Objaverse [1] have shown in the updated manuscript. We also would like to apply the model to more data domains in the future works. Please let us know if there are any additional questions that we can address.
> > > >
> > > > References:
> > > >
> > > > [1] Deitke, Matt, Dustin Schwenk, Jordi Salvador, Luca Weihs, Oscar Michel, Eli VanderBilt, Ludwig Schmidt, Kiana Ehsani, Aniruddha Kembhavi, and Ali Farhadi. "Objaverse: A universe of annotated 3d objects." In Proceedings of the IEEE/CVF Conference on Computer Vision and Pattern Recognition, pp. 13142-13153. 2023.

---

### Official Review · Reviewer_1Rvi · 2024-11-04

**Soundness:** 2
**Presentation:** 2
**Contribution:** 2
**Rating:** 5
**Confidence:** 5

**Summary:**

This paper introduces an Ambient Space Flow Transformer (ASFT), a flow-matching generative model designed to operate directly in the ambient space. The core innovation lies in eliminating the practical complexities of training latent space generative models, such as the reliance on domain-specific compressors for different data domains or the tuning of data compressor hyperparameters (e.g., adversarial weights, KL terms). Moreover, experimental results on both image and point cloud domains demonstrate competitive performance.

**Strengths:**

This paper effectively tackles the challenge of simplifying the training process for flow matching models. The proposed method is innovative and well-supported by experiments conducted on diverse datasets, including images and point clouds. The clear presentation of the problem, method, and results makes the paper easy to follow.

**Weaknesses:**

The novelty of the proposed method in this paper is questionable, as it appears to be relatively straightforward. Additionally, the experimental results are not sufficiently compelling. The use of only the ShapeNet dataset for point cloud modality limits the generalizability of the findings. More diverse datasets should be employed to validate the effectiveness of the proposed approach. Furthermore, the improvement over existing methods is not substantial, and a more comprehensive comparison with state-of-the-art methods, using a wider range of metrics, is necessary. Lastly, the visualizations in Figures 1 and 2 are unclear and do not provide a satisfactory explanation of the proposed method.

**Questions:**

The following suggestions could enhance the paper: 1) Figures 1 and 2 should be refined to provide a clearer visualization of the proposed method; 2) The experimental evaluation could be strengthened by employing a more diverse range of image and point cloud datasets to demonstrate the generalizability of the proposed approach; 3) Incorporating additional novel ideas could further enhance the distinctiveness and advancement of the proposed method.

---

> ### Author Response · Authors · 2024-11-20
> **Official Response to Reviewer 1Rvi**
>
> 1. Q: The novelty of the proposed method in this paper is questionable, as it appears to be relatively straightforward.
>     - The novelty of our work is to build a domain-agnostic generative model on coordinate-value maps (sometimes referred to as “fields”) that can be efficiently trained in large scale settings (eg. ImageNet-256). **Note that up to date,  ASFT is the only generative model in function space achieving these results. **
>    - Previous domain-agnostic generative models of fields/maps, like Functa [1], GEM [2] and DPF [3], have investigated generative models in field/function space. However, these works only tackle low-dimensional problems, like image generation of resolution 32 or 64 pixels (on smaller datasets like CelebA or LSUN-Church). In our work, we show that domain-agnostic generative models on fields/maps can achieve good performance on large datasets like ImageNet-256, which previous models fail to achieve.
>     - We want to highlight that we build ASFT with simplicity in mind and aim to build a unified generative model that can be seamlessly applied to different data domains. We are pleased that the reviewer believes that our method is straightforward, we have put significant effort to present and discuss our methodology in a way thats easy to understand.
>
> 2. Q: The experimental evaluation could be strengthened by employing a more diverse range of image and point cloud datasets to demonstrate the generalizability of the proposed approach.
>     - We evaluated our model on 3 image datasets (FFHQ, LSUN Church and ImageNet), as well, as ShapeNet for 3D shape generation. We compare with more than 20 different baselines across all these datasets and we   use all the metrics reported in previous approaches. We believe this already represents a comprehensive comparison. In addition, to demonstrate the wide applicability of ASFT, we have added results on point cloud generation of Objaverse. We train an image-to-point-cloud ASFT model on Objaverse, containing more than 800k 3D objects of wide variety. Compared with SOTA 3D generative models like CLAY, our ASFT demonstrates competitive performance on image-to-point-cloud generation. These results further demonstrate the generalizability of proposed ASFT on diverse image and point cloud generative tasks.
>         | Model | ULIP-I (↑) | P-FID (↓) |
>         | -------- | ------- | -------- |
>         | Shap-E           | 0.1307 | - |
>         | Michelangelo | 0.1899 | - |
>         | CLAY              | 0.2066 | 0.9946 |
>         | ASFT (ours)   | **0.2976** | **0.3638** |
>    - We also want to emphasize that on ShapeNet, ASFT achieves better performance than the SOTA latent diffusion model LION [1] on standard evaluation metrics, including MMD, COV, and 1-NNA. On image generation datasets FFHQ-256 and LSUN-Church-256, our model outperforms previous function-space generative models. On ImageNet-256, ASFT achieves comparable performance with models on ambient space. Admittedly, there is a performance gap compared with models trained in latent space from a pre-trained VAE. However, we want to point out that the VAE model to compute the latents is trained on a much larger dataset than ImageNet whereas ASFT is trained on ImageNet only. The widely used SD-VAE (https://huggingface.co/stabilityai/sd-vae-ft-mse) for latent space generative modeling, is trained on OpenImages (containing ~9M images) and then finetuned on subset of LAION (containing over 238k images) for a total of ~9.23M images. Whereas ImageNet contains ~1.28M images. This means that ASFT  is trained on 13% of the data used to train DiT and SiT. We have updated Tab. 7 in the appendix to reflect this fact, which we believe explains the performance gap between ASFT and latent space generative models.
>
> 3.  Q: The visualizations in Figures 1 and 2 are unclear and do not provide a satisfactory explanation of the proposed method.
>     - We have updated Figures 1 and 2 as shown to better illustrate the pipeline of proposed ASFT.

---

> > ### Author Response · Authors · 2024-11-20
> > **Official Response to Reviewer 1Rvi (2)**
> >
> > 4. Q: Incorporating additional novel ideas could further enhance the distinctiveness and advancement of the proposed method.
> >     - Our main contribution is to build a domain-agnostic generative model on coordinate-value maps (sometimes referred to as “fields”) that can be efficiently trained in large scale settings (eg. ImageNet-256), note that up to date, there’s no generative model than can do this other than ASFT.  We believe of course that there is a interesting number of questions that are natural to consider as follow up work. In particular, efficient Transformer architectures that enable even more efficient training via masking [4]
> >
> > References:
> >
> > [1] Dupont, Emilien et. al "From data to functa: Your data point is a function and you can treat it like one." Advances in Neural Information Processing Systems 2022. https://arxiv.org/pdf/2201.12204
> >
> > [2] Du, Yilun, et al. "Learning signal-agnostic manifolds of neural fields." Advances in Neural Information Processing Systems 2021. https://arxiv.org/abs/2111.06387
> >
> > [3] Zhuang, Peiye, et al. "Diffusion probabilistic fields." The Eleventh International Conference on Learning Representations. 2023.
> >
> > [4] Sehwag, Vikash, et al. "Stretching Each Dollar: Diffusion Training from Scratch on a Micro-Budget." arXiv preprint arXiv:2407.15811 (2024).

---

### Author Response · Authors · 2024-11-20
**Official Comments to All Reviewers**

We thank all reviewers for their constructive suggestions that help substantially improve the quality of our paper. We have updated our manuscript accordingly, with changes highlighted in red color. Please find below the major clarifications and updates in response to the review:

1. The main contribution of our work is to build a domain-agnostic generative model on coordinate-value maps (also referred to as “fields”) that can be efficiently trained in large scale settings (eg. ImageNet-256 and Objaverse). We aim to build a unified and simplified generative model that can be seamlessly applied to different data domains. We also want to emphasize that ASFT achieves competitive performance on standard benchmarks for image and 3D point cloud generations.  On ShapeNet, ASFT achieves better performance than the SOTA latent diffusion model LION on standard evaluation metrics. On image generation datasets FFHQ-256 and LSUN-Church-256, our model outperforms previous function-space generative models. On ImageNet-256, ASFT achieves comparable performance with models on ambient space. Admittedly, there is a performance gap compared with models trained in latent space from a pre-trained VAE. However, we want to point out that the VAE model to compute the latents is trained on a much larger dataset than ImageNet whereas ASFT is trained on ImageNet only as highlighted in the updated Tab. 3. The widely used SD-VAE (https://huggingface.co/stabilityai/sd-vae-ft-mse) for latent space generative modeling, is trained on OpenImages (containing ~9M images) and then finetuned on subset of LAION (containing over 238k images) for a total of ~9.23M images. Whereas ImageNet contains ~1.28M images. This means that ASFT is trained on 13% of the data used to train DiT and SiT. We have updated Tab. 7 in the appendix to reflect this fact, which we believe explains the performance gap between ASFT and latent space generative models.

2. We added experiments of image-to-point-cloud generation on Objaverse to validate the capability of proposed ASFT on **(1) larger and more challenging 3D generative tasks, (2) different conditioning inputs (i.e., image conditioning)**. Objaverse is a large-scale dataset that contains more than 800k 3D objects of wide variety. Results on Objaverse are listed in the table below. We report ULIP-I which measures the alignment between conditioning image and generated point clouds, as well as P-FID which measures the distribution similarity between sampled and real objects. Compared with SOTA 3D generative models like CLAY, our ASFT demonstrates strong performance on image-to-point-cloud generation. Please find more details in Appendix G and example samples in Fig. 10 of updated manuscript.
| Model | ULIP-I (↑) | P-FID (↓) |
| -------- | ------- | -------- |
| Shap-E           | 0.1307 | - |
| Michelangelo | 0.1899 | - |
| CLAY              | 0.2066 | 0.9946 |
| ASFT (ours)   | **0.2976** | **0.3638** |

3. In Fig. 11, we added more resolution agnostic sampling results. We showcase that given an ASFT trained on ImageNet-256, it can trivially generate images at high resolution from $512\times512$ to $2048\times2048$. This demonstrates the flexibility of ASFT as well as its capability to handle high-resolution in inference. Besides, we want to also highlight the results in Tab. 9 where ASFT shows better quantitative results than standard upsample strategies (e.g., bilinear and bicubic) in generating high-resolution images.

---

### Author Response · Authors · 2024-11-30
**Discussion period**

Dear SAC, AC and reviewers,

We kindly reach out to you once again to inquire about the status of the discussion. As we approach the end of the extended discussion period we want to note that only 1 out of 4 reviewers have acknowledged receiving our rebuttal. We believe our rebuttal has addressed all the points raised by reviewers (including new large scale experiments on Objaverse) and are happy to address any further suggestions from reviewers.

Thanks for your time

---

### Meta-Review · Area_Chair_qncH · 2024-12-19

**Metareview:**

The aim of the paper is to propose a domain-agnostic generative model on coordinate-value maps that can be efficiently trained at scale.
It shows show that domain-agnostic generative models on fields/maps can achieve good performance on large datasets like ImageNet-256, which previous models fail to achieve.

On the positive side, the writing is very good. The information flow is maintained, ideas are clearly explained and the presentation of the problem, method, and results makes the paper easy to follow. The proposed generative model is domain-agnostic, meaning it can be applied to image generation but can also be trained with minimal modifications for 3D point cloud generation. In the experiments, the paper shows a compelling performance at image generation, when compared to other function space approaches as well as 3D shape generation, when compared to older point-cloud generation methods.

On the negative side, I see four weaknesses that need to be addressed (listed in order of importance):

1) The paper should better work out the advantage of a domain agnostic architecture, which remains unclear. We have good domain-specific architectures for image generation and 3D generation, that perform much better compared to the proposed model. What is the advantage of the proposed domain-agnostic model? The paper mentions end-to-end optimization as the single core advantage, which might be true, but this is not worked out well in the experiments, e.g. maybe training times/ compute budgets between would be significantly different? Maybe one could actually train a domain-agnostic model that can do both image and 3D generation? Overall, I think this is a very critical point that should be discussed in more detail in the introduction and in the experiments.

2) The current results, are promising, not sufficiently convincing yet. At shape generation, the proposed model only outperforms older point-cloud based methods such as LION. The paper misses showing that it also significantly lacks behind SOTA 3D generation methods that are domain-specific, such as XCube [1] or MeshGPT [2], which I think should be included in the report. The authors argue that their lack of performance compared to SOTA image generation methods is explained by their limited training data. However, I agree with the reviewers who suggested the next step to be an attempt to scale the model and training data size to be better comparable to other methods.
[1] XCube: Large-Scale 3D Generative Modeling using Sparse Voxel Hierarchies, CVPR 2024.
[2] MeshGPT: Generating Triangle Meshes with Decoder-Only Transformers, CVPR 2024.

3) Due to the independence of the input coordinates, the paper claims a contribution allowing a very simple mechanism for sub-sampling of point clouds and super-resolution of images. However, the experiments are purely qualitative. Overall, this might be an interesting property of the architecture, but it is not convincingly presented. E.g. the images look blurry, and it remains unclear how they would compare even to the simplest upsampling methods.

In summary, the weaknesses in the presentation and experiments out-weigh the strengths of an otherwise interesting approach in a well-written paper.

**Additional Comments On Reviewer Discussion:**

3/4 reviewers were unresponsive during the discussion period with the authors as well as during the discussion with the AC. As a result, the AC made a significant effort to read through the paper, reviews and extensive rebuttal of the authors to make a well-informed decision.

The authors addressed many of the concerns of the reviewers in the rebuttal, but I think three core weaknesses remain to be addressed still (see meta review).

---

### Decision · Program_Chairs · 2025-01-22

Reject